# Relax and penalize: a new bilevel approach to mixed-binary hyperparameter optimization

**Sara Venturini**  *sventuri@mit.edu*
*MIT Senseable City Lab*
*Massachusetts Institute of Technology*

**Marianna De Santis**  *marianna.desantis@unifi.it*
*Department of Information Engineering*
*University of Florence*

**Jordan Patracone**  *jordan.patracone@univ-st-etienne.fr*
*Inria, Laboratoire Hubert Curien*
*Université Jean Monnet Saint-Etienne*[†]

**Martin Schmidt**  *martin.schmidt@uni-trier.de*
*Department of Mathematics*
*Trier University*

**Francesco Rinaldi**[*]  *rinaldi@math.unipd.it*
*Department of Mathematics*
*University of Padova*

**Saverio Salzo**[*]  *salzo@diag.uniroma1.it*
*DIAG, Sapienza University of Rome and*
*Italian Institute of Technology*

**Reviewed on OpenReview:** *https://openreview.net/forum?id=A1R1cQ93Cb*

## Abstract

In recent years, bilevel approaches have become very popular to efficiently estimate high-dimensional hyperparameters of machine learning models. However, to date, binary parameters are handled by *continuous relaxation and rounding* strategies, which could lead to inconsistent solutions. In this context, we tackle the challenging optimization of mixed-binary hyperparameters by resorting to an equivalent continuous bilevel reformulation based on an appropriate penalty term. We propose an algorithmic framework that, under suitable assumptions, is guaranteed to provide mixed-binary solutions. Moreover, the generality of the method allows to safely use existing continuous bilevel solvers within the proposed framework. We evaluate the performance of our approach for two specific machine learning problems, i.e., the estimation of the group-sparsity structure in regression problems and the data distillation problem. The reported results show that our method is competitive with state-of-the-art approaches based on relaxation and rounding.

## 1 Introduction

Nowadays, machine learning systems tend to incorporate an increasing number of hyperparameters with the purpose of improving the overall performance of learning tasks and achieving a higher flexibility. Then,

---

[*]These authors contributed equally to this work.

[†]Université Jean Monnet Saint-Etienne, CNRS, Institut d'Optique Graduate School, Inria, Laboratoire Hubert Curien UMR 5516, F-42023, SAINT- ETIENNE, France.

optimizing such high-dimensional hyperparameters becomes a a crucial step for devising an efficient and fully parameter-free machine learning systems. In recent years, bilevel approaches to hyperparameter optimization have become very popular as an effective way to estimate high-dimensional hyperparameters (Arbel & Mairal, 2022; Bae & Grosse, 2020; Bennett et al., 2006; Franceschi et al., 2018; Grazzi et al., 2020; Maclaurin et al., 2015; Pedregosa, 2016). On the other hand, in many circumstances, binary hyperparameters are included in the model to allow the pruning of the irrelevant variables or the discovery of sparsity structures. Interesting examples are given by the pruning of large-scale deep learning models (Zhang et al., 2022), the identification of the group-sparsity structures in regression problems (Frecon et al., 2018; Wang et al., 2020), and learning the discrete structure of a graph neural networks (Franceschi et al., 2019). For these cases, the usual optimization approach is that of *relaxing* the respective parameterIn over the unit interval $[0, 1]$, solve the continuous optimization problem, and then *rounding* the solution so to get a binary output. This is essentially a heuristic, which overcomes the challenge of dealing with integer variables, but does not offer any theoretical guarantees. The aim of the present work is to provide a more principled way of approaching mixed-binary hyperparameter optimization.

**Related works.** In the context of machine learning, bilevel optimization problems with binary variables arise in a number of situations. In Frecon et al. (2018) the estimation of group-sparsity structures in multi-task regression is addressed by a mixed-binary bilevel optimization model, which is handled by a continuous relaxation and approximation of the problem. The output of the optimization procedure is a vector of continuous variables, which are then rounded to the closest binary values, so to provide the final grouping of the features. In Section 5, we tackle this same problem and show the advantage of our approach. In the work Zhang et al. (2022), a new model pruning, based on bilevel optimization, is proposed, where the upper level variable is a binary mask. The related iterative algorithm performs a gradient descent step on the continuous relaxation of the problem followed by a projection step onto a discrete set, which is indeed a hard-thresholding operation. No convergence guarantees are provided. In Borsos et al. (2024), the authors present a general framework for coreset construction by formulating coreset selection as a cardinality-constrained bilevel optimization problem, solved using a tailored algorithm that combines greedy forward selection and first-order methods. The proposed approach is model-agnostic and applicable to any twice-differentiable model, including neural networks. A drawback is that the coreset weights must be determined for each selection step, which involves an iterative process. To streamline this, binary weights (i.e., unweighted coreset) are also used and a mixed-binary bilevel optimization problem is defined, thus eliminating the weight optimization step. The authors only report a theoretical analysis of the algorithm for the continuous-weights case. In another recent paper Zhou et al. (2024) some deep learning techniques are developed to tackle a bilevel problem with a binary tender, i.e., a problem where the upper and lower levels are connected through binary variables. A neural network is trained to approximate the optimal value of the lower-level problem as a function of the binary tender. This enables a single-level reformulation of the bilevel program using a mixed-integer representation of the value function. Additionally, a comparative analysis is conducted between two neural network architectures—general neural networks and novel input-supermodular neural networks—to assess their representational capacities. To handle high-dimensional bilevel programs, an enhanced sampling method is introduced to generate higher-quality samples, along with an iterative process to refine solutions.

Beyond the machine learning literature, there are a number of works related to mixed-integer bilevel programming; see, e.g., Section 5.3 of the recent survey Kleinert et al. (2021b). An important drawback one needs to take into account when applying those methods to hyperparameter optimization problems are however limited scalability. Common approaches like, e.g., the outer-approximation-based method in Kleinert et al. (2021a), or the algorithm proposed in Mitsos (2010), which requires the global solution of a a significant number of mixed-integer nonlinear programs have indeed a prohibitive cost when the dimensionality grows. Furthermore, it should be noted that they aim to achieve global optimality—an overly ambitious goal in the context of machine learning applications.

**Contributions and outline.** In this paper, we analyze more carefully the mixed-binary setting and propose a *relax and penalize* method, which produces a mixed-binary output and relies on improved mathematical grounds. More precisely, we present in Section 2 a general mixed continuous-binary bilevel problem and show in Section 3 that it is equivalent, in terms of global minima and minimizers, to a fully continuous and penalized

optimization problem. Next, in Section 4, we propose an algorithmic framework, which consists of iteratively solving a sequence of continuous and penalized problems which, under suitable assumptions, are guaranteed to provide mixed-binary local solutions. The performance of the proposed approach is quantitatively assessed on the two machine-learning applications of the group structure estimation in the group lasso problem and the data distillation task. Numerical experiments are reported in Section 5 and show how the *relax and penalize* method compares with state-of-the-art approaches based on relaxation and rounding. Finally, conclusions and perspectives are drawn in Section 6.

**Notation.** For every integer $n \geq 1$, $[n]$ denotes the set $\{1, \ldots, n\}$. We denote by $\|\cdot\|$ the Euclidean norm in $\mathbb{R}^n$ and by $\|\cdot\|_\infty$ the infinity norm, meaning $\|x\|_\infty = \max_{1 \leq u \leq n} |x_i|$. If $x \in \mathbb{R}^n$ and $\rho > 0$ we denote by $B_\rho(x)$ the closed ball in $\mathbb{R}^n$ with center $x$ and radius $\rho$, i.e., $B_\rho(x) = \{x' \in \mathbb{R}^n : \|x' - x\| \leq \rho\}$. The standard $(n-1)$-simplex is denoted by $\Delta^{n-1} = \{x \in \mathbb{R}^n_+ : \sum_{i=1}^n x_i = 1\}$. Moreover, $\odot$ is the Hadamard product, meaning the component-wise multiplication of vectors in $\mathbb{R}^d$. If $\Psi \colon \mathbb{R}^n \to \mathbb{R}$ is a continuous function and $\Omega \subseteq \mathbb{R}^n$, we denote by $\operatorname{argmin}_\Omega \Psi$ the set of minimizers of $\Psi$ over $\Omega$ and with a slight abuse of notation also the minimizer itself when it is unique.

## 2  Problem statement

We consider mixed-binary bilevel problems of the form

$$\min_{\lambda, \theta} \quad F(\lambda, \theta, w(\lambda, \theta)) \tag{1a}$$

$$\text{s.t.} \quad \lambda \in \Lambda \subseteq \mathbb{R}^m, \ \theta \in \Theta_{\text{bin}} \subseteq \{0, 1\}^p, \tag{1b}$$

$$w(\lambda, \theta) = \operatorname*{argmin}_{w \in W(\lambda, \theta)} f(\lambda, \theta, w), \tag{1c}$$

where $F, f \colon \mathbb{R}^m \times \mathbb{R}^p \times \mathbb{R}^d \to \mathbb{R}$ and $W(\lambda, \theta) \subseteq \mathbb{R}^d$. We note that the lower-level problem is supposed to admit a unique solution and that the hyperparameters $\lambda$ and $\theta$ are continuous and binary variables, respectively. In the context of machine learning problems, the functions $F$ and $f$ often are the loss over a validation set and training set, respectively. We will provide major applications of this situation in Section 5.

In the remainder of this paper we assume that the binary set $\Theta_{\text{bin}}$ is embedded in a larger continuous set $\Theta$ and that the lower-level problem admits a unique solution also for $\theta \in \Theta$. Then, we can consider the following and more compact formulation

$$\min_{(\lambda, \theta) \in \Lambda \times \Theta_{\text{bin}}} G(\lambda, \theta) \tag{2}$$

of the above problem, where we set $G(\lambda, \theta) := F(\lambda, \theta, w(\lambda, \theta))$ and require that the following assumption holds.

**Assumption 1.**

   (i) $\Lambda \subseteq \mathbb{R}^m$ *is nonempty and compact.*

   (ii) $\Theta_{\text{bin}} := \Theta \cap \{0, 1\}^p \neq \varnothing$, *where $\Theta \subseteq [0, 1]^p$ is convex and compact and $\Theta \setminus \Theta_{\text{bin}} \neq \varnothing$.*

   (iii) $G \colon \Lambda \times \Theta \to \mathbb{R}$ *is continuous.*

   (iv) *For all $\lambda \in \Lambda$, the map $G(\lambda, \cdot)$ is Lipschitz continuous with constant $L > 0$ on $\Theta$.*

Assumption 1 can be met with appropriate hypotheses on the functions $F$ and $f$. For instance, a sufficient condition for (iii) is that the functions $F$ and $f$ are jointly continuous, that the function $f(\lambda, \theta, \cdot)$ is strongly convex with a modulus of convexity which is uniform for every $(\lambda, \theta)$, and that the set-valued mapping $(\lambda, \theta) \mapsto W(\lambda, \theta)$ is closed and such that, for every $(\bar{\lambda}, \bar{\theta}) \in \Lambda \times \Theta$, $\operatorname{dist}(w(\bar{\lambda}, \bar{\theta}), W(w, \theta)) \to 0$ as $(\lambda, y) \to (\bar{\lambda}, \bar{\theta})$ (Bonnans & Shapiro, 2000, Proposition 4.4). Additional conditions can ensure the validity of (iv) too; see, Bonnans & Shapiro (2000, Section 4.4).

## 3 Restating the problem via a smooth penalty function

In order to deal with the binary variables in problem (2), we relax the integrality constraints on $\theta$ via a classic penalty term. This leads to the continuous optimization problem

$$\min_{(\lambda,\theta)\in\Lambda\times\Theta} G(\lambda,\theta) + \frac{1}{\varepsilon}\varphi(\theta) \tag{3}$$

in which we use the penalty function

$$\varphi(\theta) = \sum_{i=1}^{p} \theta_i(1-\theta_i). \tag{4}$$

Note that the function in (4) is a smooth, concave, and quadratic function with the following properties:

$$\forall\,\theta\in[0,1]^p\colon \varphi(\theta)\geq 0 \quad\text{and}\quad \forall\,\theta\in\{0,1\}^p\colon \varphi(\theta)=0.$$

This penalty has been introduced in Raghavachari (1969) to define equivalent continuous reformulations of mixed-integer linear programming problems. In Section A we give the main properties of this penalty function that we use to prove the main results of this and the next section.

We start with a result establishing the equivalence of Problems (2) and (3) in terms of global minimizers. It is in line with a stream of works analyzing the use of concave penalty functions in the framework of nonlinear optimization problems with binary or integer variables (see, e.g., Giannessi & Niccolucci (1976); Kalantari & Rosen (1987); Lucidi & Rinaldi (2010) and references therein). For the reader's convenience we provide the proof of this result in the appendix.

**Theorem 1.** *Suppose that Assumption 1 is satisfied. Then, there exists an $\bar{\varepsilon}>0$ such that for all $\varepsilon\in\,]0,\bar{\varepsilon}]$, Problems* (2) *and* (3) *have the same global minimizers, i.e.,*

$$\underset{(\lambda,\theta)\in\Lambda\times\Theta_{\mathrm{bin}}}{\operatorname{argmin}} G(\lambda,\theta) = \underset{(\lambda,\theta)\in\Lambda\times\Theta}{\operatorname{argmin}} G(\lambda,\theta) + \frac{1}{\varepsilon}\varphi(\theta).$$

The conclusion of Theorem 1 is remarkable since it guarantees that despite the fact that (3) is a purely continuous optimization problem, for $\varepsilon$ sufficiently small, all of its global minimizers are mixed-binary feasible and are exactly the global minimizers of the original problem (2).

**Remark 1.**

(i) *Set $G_\varepsilon\colon\Lambda\times\Theta\to\mathbb{R}$ such that $G_\varepsilon(\lambda,\theta)=G(\lambda,\theta)+\varepsilon^{-1}\varphi(\theta)$ and let $\delta_{\Lambda\times\Theta_{\mathrm{bin}}}\colon\Lambda\times\Theta\to\mathbb{R}$ be the indicator function of the set $\Lambda\times\Theta_{\mathrm{bin}}$, i.e., the function that is zero on $\Lambda\times\Theta_{\mathrm{bin}}$ and $+\infty$ otherwise. Then, it is easy to see that $G_\varepsilon$ $\Gamma$-converges[1] to $G+\delta_{\Lambda\times\Theta_{\mathrm{bin}}}$ as $\varepsilon\to 0$. Moreover, the family of functions $(G_\varepsilon)_{\varepsilon>0}$ is clearly equicoercive since they are all defined on the compact set $\Lambda\times\Theta$. Therefore, it holds*

$$\underset{\Lambda\times\Theta}{\operatorname{argmin}}\, G_\varepsilon \to \underset{\Lambda\times\Theta}{\operatorname{argmin}}\, G + \delta_{\Lambda\times\Theta_{\mathrm{bin}}} = \underset{\Lambda\times\Theta_{\mathrm{bin}}}{\operatorname{argmin}}\, G \quad\text{as } \varepsilon\to 0$$

*in the sense of set convergence. This is a standard result from variational analysis (Dontchev & Zolezzi, 1993) and it is always true provided that $G$ and $\varphi$ are continuous functions as well as that $\varphi\geq 0$ and $\varphi(\theta)=0$ if and only if $\theta\in\Theta_{\mathrm{bin}}$ holds.*

(ii) *In view of* (i)*, which gives an asymptotic result, the statement of Theorem 1 is stronger in the sense that, for the special function* (4) *and for $\varepsilon$ small enough, $\operatorname{argmin}_{\Lambda\times\Theta} G_\varepsilon = \operatorname{argmin}_{\Lambda\times\Theta_{\mathrm{bin}}} G$ holds.*

The previous theorem provides a justification to address problem (3) instead of (2). However, because the objective function in (3) is nonconvex, only local minimizers are computationally approachable. Thus, the idea is that of looking for local minimizers of (3) which are also mixed-binary—since the global minimizers of (2) lie among them.

The next result is entirely new and addresses the issue of identifying mixed-binary local minimizers of the objective in (3), providing a sufficient condition for that purpose.

---

[1]This type of convergence of functions is also known as epiconvergence.

**Theorem 2.** *Suppose that Assumption 1 holds. Let $c \in {]}0, 1/2[$ and $0 < \varepsilon < (1-2c)/L$. Moreover, let $(\bar{\lambda}, \bar{\theta})$ be a local minimizer of*

$$G(\lambda, \theta) + \frac{1}{\varepsilon}\varphi(\theta) \quad on \ \Lambda \times \Theta.$$

*If $\operatorname{dist}_\infty(\bar{\theta}, \Theta_{\text{bin}}) := \inf_{\theta \in \Theta_{\text{bin}}} \|\bar{\theta} - \theta\|_\infty < c$, then $\bar{\theta} \in \Theta_{\text{bin}}$.*

*Proof.* Since $\operatorname{dist}_\infty(\bar{\theta}, \Theta_{\text{bin}}) < c$, there exists $\theta \in \Theta_{\text{bin}}$ such that $\|\bar{\theta} - \theta\|_\infty < c$. Let

$$\theta_t := (1-t)\bar{\theta} + t\theta = \bar{\theta} + t(\theta - \bar{\theta}) \quad \text{with } t \in [0, 1].$$

In particular, $\|\theta_t - \bar{\theta}\|_\infty = t\|\theta - \bar{\theta}\|_\infty < tc \le c$ and $\theta_t \in \Theta$, since $\Theta$ is convex. By Lemma 3,

$$\varphi(\bar{\theta}) - \varphi(\theta_t) \ge (1-2c)\|\theta_t - \bar{\theta}\| \tag{5}$$

holds. Moreover, there exists $\rho > 0$ such that for all $(\lambda', \theta') \in B_\rho(\bar{\lambda}, \bar{\theta}) \cap (\Lambda \times \Theta)$, it holds

$$G(\bar{\lambda}, \bar{\theta}) + \frac{1}{\varepsilon}\varphi(\bar{\theta}) \le G(\lambda', \theta') + \frac{1}{\varepsilon}\varphi(\theta').$$

Now, take $t \in {]}0, 1[$ such that $t < \rho/(c\sqrt{p})$. Then, $\theta_t \in \Theta$ and $\|\theta_t - \bar{\theta}\| \le \sqrt{p}\|\theta_t - \bar{\theta}\|_\infty < \sqrt{p}\, tc < \rho$. Therefore, $(\bar{\lambda}, \theta_t) \in B_\rho(\bar{\lambda}, \bar{\theta}) \cap (\Lambda \times \Theta)$ and we obtain

$$\begin{aligned}
G(\bar{\lambda}, \theta_t) - G(\bar{\lambda}, \bar{\theta}) + \frac{1}{\varepsilon}\varphi(\theta_t) - \frac{1}{\varepsilon}\varphi(\bar{\theta}) &\le L\|\theta_t - \bar{\theta}\| + \frac{1}{\varepsilon}(\varphi(\theta_t) - \varphi(\bar{\theta})) \\
&\overset{(5)}{\le} L\|\theta_t - \bar{\theta}\| - \frac{1-2c}{\varepsilon}\|\theta_t - \bar{\theta}\| \\
&= \underbrace{\left(L - \frac{1-2c}{\varepsilon}\right)}_{<0}\|\theta_t - \bar{\theta}\|. \tag{6}
\end{aligned}$$

Moreover, if $\bar{\theta} \notin \{0, 1\}^p$, since $\theta \in \{0, 1\}^p$, we have $\|\theta - \bar{\theta}\| > 0$ and hence $\|\theta_t - \bar{\theta}\| > 0$ for $t > 0$. Thus, (6) is strictly negative and it holds

$$G(\bar{\lambda}, \theta_t) + \frac{1}{\varepsilon}\varphi(\theta_t) < G(\bar{\lambda}, \bar{\theta}) + \frac{1}{\varepsilon}\varphi(\bar{\theta}),$$

which gives a contradiction. Thus, necessarily $\bar{\theta} \in \{0, 1\}^p$. $\qquad\square$

**Remark 2.**

(i) *Theorem 2 essentially says that, if $\varepsilon$ is small enough, within the distance of $1/2$ measured with the infinity norm, there are no other local minimizers of (3) than the ones that are mixed-binary feasible.*

(ii) *Note that it does not make much sense to consider local minimizers of the function $G$ over $\Lambda \times \Theta_{\text{bin}}$, since any point in $\Theta_{\text{bin}}$ is an isolated point and thus one can find a corresponding local minimizer for each one of them.*

## 4 An iterative penalty method

We now present an iterative method addressing problem (2). The idea is that of solving a sequence of problems of the form (3), indexed with $k$, with decreasing parameters $\varepsilon_k$. Hence, the problem to be solved in each iteration reads

$$\min_{(\lambda, \theta) \in \Lambda \times \Theta} G(\lambda, \theta) + \frac{1}{\varepsilon^k}\varphi(\theta). \tag{$\text{P}^k$}$$

Then, thanks to Theorem 1, it is clear that after a finite number of iterations, the original mixed-binary optimization problem and the relaxed and penalized one ($\text{P}^k$) become equivalent in terms of global minimizers. Moreover, as we have already discussed in the previous section, in practice we can only target the computation of local minimizers, but we can restrict the search to the mixed-binary ones. In the following, we make this strategy more precise.

---

**Algorithm 1:** Penalty method

---

**Input:** Problem (2), $\varepsilon^0 > 0$, $\beta \in \;]0,1[$.

**1 for** $k = 0, 1, 2, \dots$ **do**

**2**     Let $(\lambda^k, \theta^k)$ be a solution (either local or global) of problem $(\mathrm{P}^k)$.

**3**     **if** $\theta^k \notin \{0,1\}^p$ **then**

**4**        update $\varepsilon^{k+1} = \beta \varepsilon^k$

**5**     **else**

**6**        **return** $(\lambda^k, \theta^k)$.

---

**Theorem 3.** *Suppose that Assumption 1 holds. Let $(\varepsilon_k)_{k \in \mathbb{N}}$ be a vanishing sequence of positive numbers and, for every $k \in \mathbb{N}$, let $(\lambda^k, \theta^k)$ be a local minimizer of $(\mathrm{P}^k)$. Then,*

$$\liminf_{k \to +\infty} \mathrm{dist}_\infty(\theta^k, \Theta_{\mathrm{bin}}) < 1/2 \implies \exists\, k \in \mathbb{N} \; s.t. \; \theta^k \in \Theta_{\mathrm{bin}}.$$

*Moreover, if $\theta^k \in \Theta_{\mathrm{bin}}$, then we have that $\lambda^k$ is a local minimizer of*

$$\min_{\lambda \in \Lambda} G(\lambda, \theta^k).$$

*Proof.* Suppose that $\liminf\limits_{k \to +\infty} \mathrm{dist}_\infty(\theta^k, \Theta_{\mathrm{bin}}) < 1/2$ and let $c > 0$ such that $\liminf\limits_{k \to +\infty} \mathrm{dist}_\infty(\theta^k, \Theta_{\mathrm{bin}}) < c < 1/2$. Then, there exists a subsequence $(\theta^{n_k})_{k \in \mathbb{N}}$ such that

$$\forall\, k \in \mathbb{N}: \quad \mathrm{dist}_\infty(\theta^{n_k}, \Theta_{\mathrm{bin}}) < c \quad \text{and} \quad \varepsilon^{n_k} \to 0.$$

Thus, there exists $k \in \mathbb{N}$ such that

$$\mathrm{dist}_\infty(\theta^{n_k}, \Theta_{\mathrm{bin}}) < c \quad \text{and} \quad \varepsilon_{n_k} < \frac{1 - 2c}{L}$$

and this, in view of Theorem 2, gives that $\theta^{n_k} \in \Theta_{\mathrm{bin}}$. Concerning the second part of the statement, suppose that $\theta^k \in \Theta_{\mathrm{bin}}$, where $(\lambda^k, \theta^k)$ is a local minimizer of $(\mathrm{P}^k)$. Then, $\theta^k \in \{0,1\}^p$ and there exists $\rho_k > 0$ such that

$$\forall\, (\lambda, \theta) \in B_{\rho_k}(\lambda^k, \theta^k) \cap (\Lambda \times \Theta): \; G(\lambda^k, \theta^k) + \frac{1}{\varepsilon_k}\varphi(\theta^k) \le G(\lambda, \theta) + \frac{1}{\varepsilon_k}\varphi(\theta).$$

Therefore, taking $\theta = \theta^k$ in the above inequality and noting that $\varphi(\theta^k) = 0$, we have

$$\forall\, \lambda \in B_{\rho_k}(\lambda^k) \cap \Lambda: \; G(\lambda^k, \theta^k) + \frac{1}{\varepsilon_k}\underbrace{\varphi(\theta^k)}_{=0} \le G(\lambda, \theta^k) + \frac{1}{\varepsilon_k}\underbrace{\varphi(\theta^k)}_{=0},$$

which shows that $\lambda^k$ is a local minimizer of $G(\cdot, \theta^k)$ over $\Lambda$. $\qquad\square$

**Remark 3.**

(i) *In the experiments given in Section 5, we checked that the condition considered in Theorem 3 always occurs, meaning that the distance $\mathrm{dist}_\infty(\theta^k, \Theta_{\mathrm{bin}})$, where $\theta^k$ was obtained by solving problem $(\mathrm{P}^k)$ via a gradient-based subroutine, remains well-below the threshold $1/2$ for $k$ sufficiently large (See Appendix D.3 for a plot of the evolution along the iterations).*

(ii) *We note that there might indeed exist points $\theta \in \Theta$ such that $\mathrm{dist}_\infty(\theta, \Theta_{\mathrm{bin}}) > 1/2$. For instance, if we take the standard $(p-1)$-simplex*

$$\Theta = \Delta^{p-1} = \left\{ \theta \in \mathbb{R}^p_+ : \sum_{i=1}^{p} \theta_i = 1 \right\},$$

*we have that*

$$\{\theta \in \Theta : \text{dist}_\infty(\theta, \Theta_{\text{bin}}) \geq 1/2\} = \Delta^{p-1} \cap [0, 1/2]^p, \tag{7}$$

*which for $p = 3$ is the full equilateral triangle with vertices $(e_1 + e_2)/2, (e_1 + e_3)/2$ and $(e_2 + e_3)/2$, where the $e_i$'s are the vectors of the canonical basis of $\mathbb{R}^p$. In general, the set in (7) is a polytope of dimension $p - 1$ with $2p$ facets and $(p(p-1)/2)$ vertices.*

The method is formally given in Algorithm 1.

## 5 Two machine-learning applications

In this section, we present two machine-learning applications: estimating the group-sparsity structure in regression problems (Subsection 5.1) and performing data distillation (Subsection 5.2). Firstly, we present the problem setting and the bilevel formulation. Secondly, we show how the problem fits our mathematical formulation. Finally, we provide a comparison with relaxation and rounding using two different rounding strategies: the *simple rounding* baseline, which obtains the mixed-binary solution by rounding each variable individually to 0 or 1, and the *top-k hard thresholding baseline*, where the mixed-integer solution is obtained by performing a top-$k$ hard thresholding operation on the continuous solution [2] (Frecon et al., 2018; Zhang et al., 2022). The codes are available on the GitHub page: `https://github.com/saraventurini/Relax-and-penalize`

### 5.1 Group lasso structure

Here, we present the application of estimating group-sparsity structures, which is useful in areas such as gene expression analysis. We follow the formulation, the optimization algorithm, and the experimental setup in Frecon et al. (2018). In particular, we extend the approach by optimizing over both the hyperparameters $\theta$ and $\lambda$, instead of determining $\lambda$ by cross-validation. We compare our *relax and penalize* strategy with the *relaxation and rounding* proposed in Frecon et al. (2018), using both *simple* and *top-k hard thresholding* rounding. The datasets used in the tests are challenging variants of the synthetic datasets referenced in Frecon et al. (2018), specifically designed to create classes of differing sizes.

**Problem setting and formulation.** Given an output vector $y \in \mathbb{R}^N$ and a design matrix $X \in \mathbb{R}^{N \times P}$, the group lasso problem can be formulated as follows

$$\min_{w \in \mathbb{R}^P} \frac{1}{2} \|Xw - y\|^2 + \lambda \sum_{l=1}^{L} \|\theta_l \odot w\|_2,$$

where $\lambda > 0$ is a regularization parameter and $\theta_l$ is a binary vector (with entries in $\{0, 1\}$) indicating the features (components) of $w$ belonging to the $l$th group, meaning $\mathcal{G}_l = \{i \in [P] : \theta_{i,l} = 1\}$, where the vectors $\theta_l$ are thought as columns of a $P \times L$ matrix. In the case of nonoverlapping groups, it is assumed that $\sum_{l=1}^{L} \theta_{j,l} = 1$ for every $j \in [P]$. In the classic literature on the topic, the groups are assumed to be known a priori (Yuan & Lin, 2006; Zhao et al., 2009), but often in practice there is no clue about the structure of the groups and the problem is to infer this group structure from the data. However, this amounts to estimating the binary variables $\theta_l$'s, which in general poses a challenge.

In view of the discussion above, in the related literature a common approach is to relax the problem allowing the $\theta_l$'s to vary in the continuum $[0, 1]^P$. This approach was followed in Frecon et al. (2018), in which the following bilevel optimization problem is proposed

$$\min_{\theta \in \Theta} \frac{1}{T} \sum_{t=1}^{T} C_t(w_t(\lambda, \theta)) \quad \text{with} \quad \Theta = \left\{ \theta \in [0, 1]^{P \times L} : \sum_{l=1}^{L} \theta_l = \mathbf{1}_P \right\} = (\Delta^{L-1})^P$$

$$\text{and} \quad w(\lambda, \theta) = \underset{(w_1, \dots, w_T) \in \mathbb{R}^{d \times T}}{\text{argmin}} \frac{1}{T} \sum_{t=1}^{T} \left( \frac{1}{2} \|X_t w_t - y_t\|^2 + \lambda \sum_{l=1}^{L} \|\theta_l \odot w_t\|_2 + \frac{\eta}{2} \|w_t\|^2 \right). \tag{8}$$

---

[2] which can be equivalently defined as rounding up to 1 the largest $k$ components of the vector and 0 the other ones.

Here, $(X_t, y_t)_{1 \leq t \leq T}$ defines $T$ regression problems in which the regressors share the same group-sparsity structure, $C_t$ is a smooth cost function acting as a validation error for the $t$-th task, and $\theta_l$ are thought as columns of a $P \times L$ matrix. Note that in this formulation, $\lambda$ is supposed to be fixed (possibly determined by a cross-validation procedure). The regularization terms $\eta/2\|w_t\|^2$, with $\eta \ll 1$, are added to ensure uniqueness of the solution to the lower-level problem and to devise a dual algorithmic procedure generating a sequence $(w^{(q)}(\lambda, \theta))_{q \in \mathbb{N}}$ with smooth updates (w.r.t. $\theta$) such that $w^{(q)}(\lambda, \theta) \to w(\lambda, \theta)$ uniformly on $\Theta$ as $q \to +\infty$; see Frecon et al. (2018, Section 3.2). Ultimately, the groups are estimated by solving the problem

$$\min_{\theta \in \Theta} \frac{1}{T} \sum_{t=1}^{T} C_t(w^{(q)}(\lambda, \theta)),$$

with $q$ large enough, and by appropriately thresholding (a posterior) the solution $\theta$ in order to recover binary variables $\theta_l$'s.

**Proposed method.** Our general *relax and penalize* approach, as described in Section 4, allows us to bypass the last thresholding step and directly address the more challenging problem

$$\min_{(\lambda,\theta) \in \Lambda \times \Theta_{\text{bin}}} \frac{1}{T} \sum_{t=1}^{T} C_t(w^{(q)}(\lambda, \theta)) \quad \text{with} \quad \begin{cases} \Lambda = [\lambda_{\min}, \lambda_{\max}] & \text{with } 0 < \lambda_{\min} < \lambda_{\max}, \\ \Theta_{\text{bin}} = \left\{ \theta \in \{0,1\}^{P \times L} : \forall j \in [P] \ \sum_{l=1}^{L} \theta_{j,l} = 1 \right\}, \end{cases} \tag{9}$$

Note that in Frecon et al. (2018), $\lambda$ is supposed to be fixed (possibly determined by a cross-validation procedure). Instead, here we are optimizing w.r.t. both the hyperparameters $\theta$ and $\lambda$, obtaining a mixed-binary problem in the end. In Section C we report the details of the extension. Now, since $w^{(q)}(\lambda, \theta)$ is smooth w.r.t. $\theta$, the objective in (9) satisfies Assumption 1 and hence, in view of Theorem 1 and Theorem 3, we can consider the *relaxed and penalized* version of Problem (9), that is

$$\min_{(\lambda,\theta) \in \Lambda \times \Theta} \frac{1}{T} \sum_{t=1}^{T} \left( C_t(w^{(q)}(\lambda, \theta)) + \frac{1}{\varepsilon} \varphi(\theta) \right), \tag{10}$$

and state that, if $\varepsilon$ is small enough, the two problems (9) and (10) share the same global minimizers. By leveraging this equivalence, we study in the next section the added benefits of the proposed Algorithm 1.

**Remark 4.** *According to Frecon et al. (2018, Theorem 3.1) we have that $w^{(q)}(\lambda, \theta) \to w(\lambda, \theta)$ as $q \to +\infty$ uniformly on $\Lambda \times \Theta_{\text{bin}}$, so that, similarly to Frecon et al. (2018, Theorem 2.1), one can prove that Problem (9) converges as $q \to +\infty$ to the problem*

$$\min_{(\lambda,\theta) \in \Lambda \times \Theta_{\text{bin}}} \frac{1}{T} \sum_{t=1}^{T} C_t(w(\lambda, \theta))$$

*in terms of optimal values and sets of global minimizers. This provides a justification for addressing Problem (9).*

**Numerical experiments.** The experimental setting is similar to that of Frecon et al. (2018). Indeed, we create synthetic datasets where each task amounts to predict an oracle regressor $w^\star \in \mathbb{R}^P$ and an oracle group structure $\theta^\star \in \mathbb{R}^{P \times L}$, such that for all $j \in [P]$ and $l \in [L]$, $\theta_{j,l}^\star = 1$ if $j$ belongs to group $l$ and 0 otherwise. We consider $P = 100$ features, $N = 20$ observations, $T = 500$ tasks, and $L = 10$ oracle groups. For every task $t \in [T]$, we generate the oracle regressor $w_t^\star$ and the data $(X_t, y_t)$ as follows. The regressor $w_t^\star$ is generated such that its values are non-zero in one group chosen at random. In particular, we study the datasets as these values belong to the interval $[-1, -a] \cup [a, 1]$ as $0 < a \leq 1$ varies. We point out that in Frecon et al. (2018), $w_t^\star$ are considered binary, and the smaller $a$ is, the more difficult the problem becomes. The design matrix $X_t \in \mathbb{R}^{N \times P}$ is randomly drawn according to a standard normal distribution and then normalized column-wise. The output $y_t$ is such that $y_t \sim \mathcal{N}(X_t w_t^\star, 0.2 \text{I}_D)$. Validation and test sets are generated similarly. Unlike Frecon et al. (2018), we allow groups of completely random sizes, with no

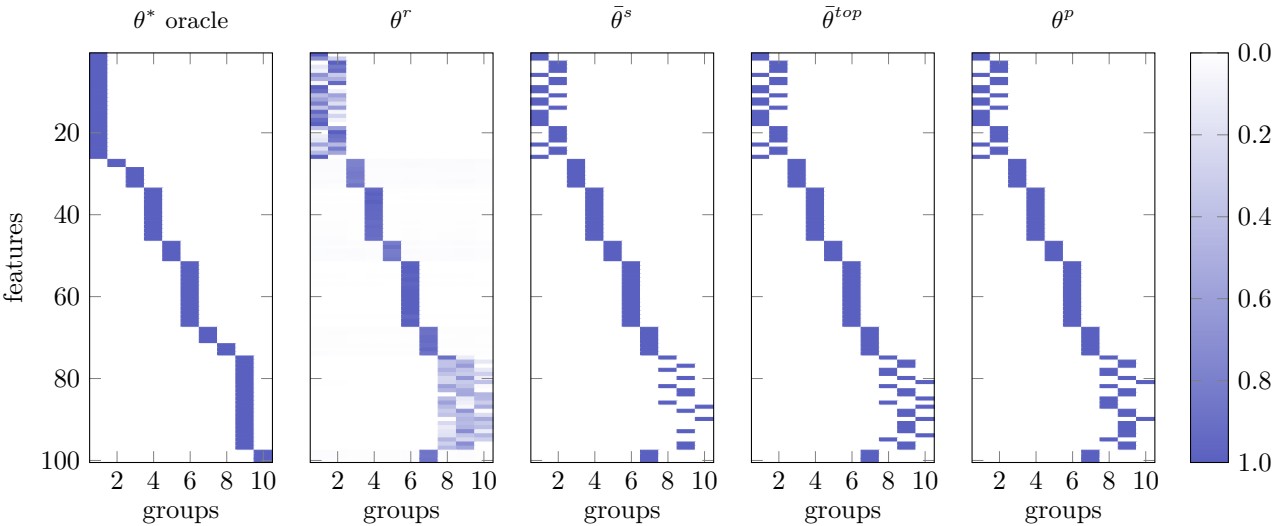

Figure 1: Example of oracle group structure with *random* sizes and parameter $a = 0.5$, and the corrisponding $\theta$ obtained by the *relaxation and rounding* before the rounding procedure (ref. as $r$), after both *simple* (ref. as $s$) and *top-1 hard thresholding* rounding (ref. as *top*), and the *relax and penalize* (ref. as $p$) methods.

Table 1: Test errors (mean ± standard deviation), i.e., the value of the upper level functions $G$, for the *relaxation and rounding* (ref. as $r$), with both *simple* (ref. as $s$) and *top-1 hard thresholding* rounding (ref. as *top*), and the *relax and penalize* (ref. as $p$) methods, over 3 runs of *inequal* and *random* group structures, for $a \in \{0.1, 0.3, 0.5\}$.

| | INEQUAL | | | RANDOM | | |
|---|---|---|---|---|---|---|
| | 0.1 | 0.3 | 0.5 | 0.1 | 0.3 | 0.5 |
| $G(\lambda^r, \theta^\star)$ | $0.04 \pm 0.00$ | $0.05 \pm 0.00$ | $0.06 \pm 0.00$ | $0.06 \pm 0.00$ | $0.07 \pm 0.00$ | $0.09 \pm 0.00$ |
| $G(\lambda^r, \theta^r)$ | $0.04 \pm 0.00$ | $0.05 \pm 0.00$ | $0.06 \pm 0.00$ | $0.05 \pm 0.00$ | $0.06 \pm 0.00$ | $0.07 \pm 0.01$ |
| $G(\lambda^r, \bar{\theta}^s)$ | $0.06 \pm 0.02$ | $0.05 \pm 0.00$ | $0.06 \pm 0.00$ | $0.12 \pm 0.05$ | $0.09 \pm 0.03$ | $0.14 \pm 0.06$ |
| $G(\lambda^r, \bar{\theta}^{\mathrm{top}})$ | $\mathbf{0.04 \pm 0.00}$ | $\mathbf{0.05 \pm 0.00}$ | $\mathbf{0.06 \pm 0.00}$ | $\mathbf{0.05 \pm 0.00}$ | $\mathbf{0.06 \pm 0.01}$ | $\mathbf{0.07 \pm 0.01}$ |
| $G(\lambda^p, \theta^\star)$ | $0.04 \pm 0.00$ | $0.05 \pm 0.00$ | $0.06 \pm 0.01$ | $0.06 \pm 0.00$ | $0.07 \pm 0.00$ | $0.09 \pm 0.00$ |
| $G(\lambda^p, \theta^p)$ | $0.05 \pm 0.00$ | $\mathbf{0.05 \pm 0.00}$ | $\mathbf{0.06 \pm 0.01}$ | $\mathbf{0.05 \pm 0.00}$ | $\mathbf{0.06 \pm 0.01}$ | $\mathbf{0.07 \pm 0.01}$ |

predefined criteria. In particular, we consider two settings: the *unequal* one, with half groups of size 5 and half groups of size 15, and the *random* one, where the size of the groups are allowed to be different and are calculated as follow. We generate $L$ normally distributed random values, apply the softmax operation to obtain $L$ percentages, and use these percentages to distribute the $P$ features across the different groups. All the results are averaged over 5 runs. See Section C for more details.

We compare several methods, which produce the following quantities: $(\lambda^r, \theta^r)$ obtained by *a pure relaxation* method, without any rounding, $(\lambda^r, \bar{\theta}^s)$ obtained by a *simple* rounding of the relaxed solution, $(\lambda^r, \bar{\theta}^{\mathrm{top}})$ obtained by *top-1 hard thresholding* the relaxed solution, meaning assigning each feature to its dominant group, and finally $(\lambda^p, \theta^p)$ obtained by the proposed *relax and penalize* method. In Figure 1 we first show an example of the different group structures retrieved by the various methods compared to the oracle group structure $\theta^\star$, for a *random* synthetic dataset with $a = 0.5$. We can notice that using directly the *relaxed* solution, some values are less than 0.5. As a result, after a simple rounding, some rows may end up as zeros, leaving those features unassigned to any group. Then, we evaluate the methods using two performance measures, which are also evaluated at the oracle group structure $\theta^\star$. The related results are reported in Table 1 and Table 2. Notice that $\theta^\star$, $\bar{\theta}^s$, $\bar{\theta}^{\mathrm{top}}$, $\theta^p$ are binary instead $\theta^r$ might not be so. Overall, these preliminary experiments on a group lasso problem show that the proposed method and the *relaxation and rounding method* with top-1 hard thresholding perform essentially the same, and they are both preferable to

Table 2: Reconstruction errors (mean $\pm$ standard deviation), i.e., Frobenius norm of the difference of the oracle regressor and the obtained regressors, for the *relaxation and rounding* (ref. as $r$), with both *simple* (ref. as $s$) and *top-1 hard thresholding* rounding (ref. as *top*), and the *relax and penalize* (ref. as $p$) methods, over 3 runs of *inequal* and *random* group structures, for $a \in \{0.1, 0.3, 0.5\}$.

| | INEQUAL | | | RANDOM | | |
|---|---|---|---|---|---|---|
| | 0.1 | 0.3 | 0.5 | 0.1 | 0.3 | 0.5 |
| $\|w(\lambda^r, \theta^\star) - w^\star\|_F$ | $4.20 \pm 0.21$ | $5.24 \pm 0.19$ | $4.67 \pm 0.15$ | $5.34 \pm 0.18$ | $5.96 \pm 0.16$ | $6.69 \pm 0.15$ |
| $\|w(\lambda^r, \theta^r) - w^\star\|_F$ | $4.73 \pm 0.35$ | $5.64 \pm 0.05$ | $5.13 \pm 0.22$ | $5.50 \pm 0.24$ | $6.10 \pm 0.25$ | $6.86 \pm 0.19$ |
| $\|w(\lambda^r, \bar{\theta}^s) - w^\star\|_F$ | $7.93 \pm 1.52$ | $5.99 \pm 1.17$ | $6.63 \pm 0.16$ | $11.17 \pm 2.51$ | $9.17 \pm 2.05$ | $11.95 \pm 2.72$ |
| $\|w(\lambda^r, \bar{\theta}^{\mathrm{top}}) - w^\star\|_F$ | $\mathbf{5.10 \pm 0.37}$ | $\mathbf{5.99 \pm 0.15}$ | $\mathbf{5.40 \pm 0.16}$ | $\mathbf{5.93 \pm 0.16}$ | $\mathbf{6.50 \pm 0.41}$ | $7.18 \pm 0.25$ |
| $\|w(\lambda^p, \theta^\star) - w^\star\|_F$ | $4.20 \pm 0.20$ | $5.32 \pm 0.20$ | $4.69 \pm 0.15$ | $5.33 \pm 0.16$ | $5.92 \pm 0.11$ | $6.75 \pm 0.12$ |
| $\|w(\lambda^p, \theta^p) - w^\star\|_F$ | $5.15 \pm 0.29$ | $6.04 \pm 0.11$ | $5.54 \pm 0.18$ | $5.94 \pm 0.17$ | $6.57 \pm 0.39$ | $\mathbf{7.17 \pm 0.23}$ |

the relaxed and simple rounding procedure. We believe that the effectiveness of the top-1 hard thresholding is due the the fact that we are considering nonoverlapping group structures which put strong constraints on the binary variables, since it is all about identifying one nonzero entry for each feature. Finally, it is worth noting that the group lasso problems used in the study are randomly generated synthetic problems. As such, they appear to be less challenging compared to the real-world problems addressed in the next experiment.

## 5.2 Data distillation

We now present the application of data distillation. Data distillation is a process that synthesizes compact summaries of large datasets, enabling efficient model training and inference. Preserving essential information while reducing size allows quicker processing and improved performance in various applications. In Sachdeva & McAuley (2023), the authors give an extensive survey on data distillation and propose a bilevel optimization model to handle the task (see Section 2 in Sachdeva & McAuley (2023)), which is the same model described here. We optimize the lower-level problem exactly, and the upper-level problem with a stochastic projected gradient descent. We test the *relaxation and rounding*, with both *simple* and *top-k hard thresholding* rounding (Zhang et al., 2022), and *relax and penalize* strategies over two real datasets, showing the effectiveness of the last one.

**Problem setting and formulation.** Given a training dataset that needs to be distilled $\mathcal{D}^{\mathrm{train}} = \{(x_i^{\mathrm{train}}, y_i^{\mathrm{train}})\}_{i=1}^m$ and a data budget $\tau \in \mathbb{Z}^+$, data distillation techniques aim to synthesize a high-fidelity data summary $\mathcal{D}_{\mathrm{syn}}^{\mathrm{train}} = \{(x_i^{\mathrm{train}}, y_i^{\mathrm{train}})\}_{i=1}^\tau$ with $\tau \ll m$. Given a validation set $\mathcal{D}^{\mathrm{val}} = \{(x_j^{\mathrm{val}}, y_j^{\mathrm{val}})\}_{j=1}^n$, the data distillation task can be formulated as the following bilevel optimization problem

$$\min_{v \in \{0,1\}^m, e^\top v = \tau} \ell^{\mathrm{val}}(\zeta(v)) \quad \text{with} \quad \zeta(v) = \operatorname*{argmin}_{\zeta} \ell^{\mathrm{train}}(\zeta, v) + s\|\zeta\|^2 \tag{11}$$

with validation loss $\ell^{\mathrm{val}}$, weighted training loss $\ell^{\mathrm{train}} = \sum_{i=1}^m v_i \ell(x_i^{\mathrm{train}}, y_i^{\mathrm{train}}, \zeta)$, regularization parameter $s \in \mathbb{R}$, $v$ being a binary vector of dimension $m$ indicating the samples in $\mathcal{D}_{\mathrm{syn}}^{\mathrm{train}}$, and $e^\top v = \tau$ is the knapsack constraint to take into account the budget. Therefore, we wish to have $v_i = 1$ for the $\tau$ most representative samples.

A simple approach to solve Problem (11) is to relax it by allowing the $v_i$'s to vary within the interval $[0, 1]$, projecting them to the region defined by the knapsack constraint, and then rounding them at the end. In particular, in our case we suppose $x_i, x_j \in \mathbb{R}^d$ as well as $y_i, y_j \in \mathbb{R}^e$ and we consider $\zeta = (W, b)$ a linear model with weight $W$ and bias $b$ as well as $\ell^{\mathrm{val}}$ and $\ell$ being mean squared error losses. Therefore, the aim is to solve the bilevel integer problem

$$\min_{v \in [0,1]^m, e^\top v = \tau} \frac{1}{2n} \sum_{j=1}^n \|W(v)x_j^{\mathrm{val}} + b(v) - y_j^{\mathrm{val}}\|^2$$

$$\text{with} \quad (W(v), b(v)) = \operatorname*{argmin}_{W \in \mathbb{R}^{e \times d}, b \in \mathbb{R}^e} \frac{1}{m} \sum_{i=1}^m v_i \|W x_i^{\mathrm{train}} + b - y_i^{\mathrm{train}}\|^2 + s\|W\|^2. \tag{12}$$

We indicate with $\Theta = \{v \in [0,1]^m : e^\top v = r\}$, and with $\Theta_{\text{bin}} = \Theta \cap \{0,1\}^m$.

**Proposed method.**   Using the approach presented in Section 4, we avoid the final thresholding and directly optimize the following problem

$$\min_{v \in \{0,1\}^m, e^\top v = \tau} \frac{1}{2n} \sum_{j=1}^n \|W(v) x_j^{\text{val}} + b(v) - y_j^{\text{val}}\|^2$$

$$\text{with} \quad (W(v), b(v)) = \underset{W \in \mathbb{R}^{e \times d}, b \in \mathbb{R}^e}{\operatorname{argmin}} \frac{1}{m} \sum_{i=1}^m v_i \|W x_i^{\text{train}} - b - y_i^{\text{train}}\|^2 + s\|W\|^2. \tag{13}$$

We solve the lower-level problem exactly, obtaining

$$W(v) = \left(\frac{1}{m} C_v(X,Y)\right) \left(\frac{1}{m} C_v(X) + s I_d\right)^{-1}, \quad b(v) = \bar{y}_v - W(v) \bar{x}_v, \tag{14}$$

with

$$C_v(X,Y) = \sum_{i=1}^m v_i (y_i^{\text{train}} - \bar{y}_v)(x_i^{\text{train}} - \bar{x}_v)^\top$$

being the cross-covariance matrix,

$$C_v(X) = \sum_{i=1}^m v_i (x_i^{\text{train}} - \bar{x}_v)(x_i^{\text{train}} - \bar{x}_v)^\top$$

being the covariance matrix, and

$$\bar{x}_v = \frac{1}{\sum_{i=1}^m v_i} \sum_{i=1}^m v_i x_i, \quad \bar{y}_v = \frac{1}{\sum_{i=1}^m v_i} \sum_{i=1}^m v_i y_i.$$

All the results are averaged over 5 runs. See Section D for more details.

We can also write the problem as

$$\min_v \ell^{\text{val}}(\zeta(W(v), b(v))) \quad \text{with} \quad v \in \{0,1\}^m, \; e^\top v = \tau. \tag{15}$$

We solve the upper-level problem by means of a projected stochastic gradient descent method. To efficiently perform the projection over the knapsack constraint, we use the Kiwiel algorithm (Kiwiel, 2008); see Section D for further details. Now, since $W(v)$ and $b(v)$ are smooth w.r.t. $v$, the objective in (15) satisfies Assumption 1 and hence, in view of Theorem 1 and Theorem 3, we can consider the *relaxed and penalized* version of Problem (15), i.e.,

$$\min_v \ell^{\text{val}}(\zeta(w(v), b(v))) + \frac{1}{\varepsilon} \varphi(v) \quad \text{with} \quad v \in [0,1]^m, \; e^\top v = \tau \tag{16}$$

and state that, if $\varepsilon$ is small enough, the two problems (15) and (16) share the same global minimizers.

**Numerical experiments.**   We tested the proposed method on the data distillation problem by performing experiments on two real-world datasets. First, *music* (Bertin-Mahieux, 2011), a dataset with song features from 1922 to 2011 used to predict the release year based on 90 attributes, including timbre averages and covariances. Second, *blog* (Buza, 2014), a dataset containing features from blog posts, focused on predicting the number of comments received in the next 24 hours using various attributes. For each dataset, we address Problem (12) by using a training set for the lower level and a validation set for the upper level. We vary the distillation budget $\tau$ at 10 %, 20 %, 30 % of the training set size; see Section D for more details. We run the experiments with the *relaxation and rounding*, with both *simple* and *top-k hard thresholding* rounding, and the *relax and penalize* strategies, and we report the results in Table 3. First, we compare the final objective values of the upper-level problem obtained from *relaxation and rounding*, with both rounding strategies, alongside those from *relax and penalize*, including the actual number of training samples selected in $D_{\text{syn}}^{\text{train}}$. Next, we

Table 3: Experiments on the regression task across two real-world datasets: *music* and *blog*. The distillation budget $\tau$ is varied at $10\%$, $20\%$, and $30\%$ (*perc* column) of the training set size. For the *relaxation and rounding*, with both *simple* (simple) and *top-k hard thresholding* rounding (top-k), and the *relax and penalize* (*penalize*) methods, we report (mean $\pm$ standard deviation over 5 runs) the values of the upper level function (before rounding in brackets) for the validation and test sets, the cardinality of the synthesized set, and the RMSE.

| dataset | perc | $\tau$ | method | $\ell^{\text{val}}$ | $\left\lvert D_{\text{syn}}^{\text{train}} \right\rvert$ | $\ell^{\text{test}}$ | RMSE |
|---|---|---|---|---|---|---|---|
| *music* | 10 | 23186 | simple | $126.85 \pm 1.37\ (58.57 \pm 0.02)$ | $12764.20 \pm 28.63$ | $128.61 \pm 1.37$ | $16.04 \pm 0.09$ |
| | | | top-k | $65.60 \pm 0.32\ (58.57 \pm 0.02)$ | $23186.00 \pm 0.00$ | $67.02 \pm 0.30$ | $11.58 \pm 0.03$ |
| | | | penalize | $\mathbf{64.87 \pm 0.70}$ | $\mathbf{23186.00 \pm 0.00}$ | $\mathbf{65.96 \pm 0.08}$ | $\mathbf{11.49 \pm 0.01}$ |
| | 20 | 46371 | simple | $84.84 \pm 0.40\ (53.37 \pm 0.01)$ | $24967.00 \pm 47.02$ | $86.19 \pm 0.41$ | $13.13 \pm 0.03$ |
| | | | top-k | $54.61 \pm 0.02\ (53.37 \pm 0.01)$ | $46371.00 \pm 0.00$ | $55.82 \pm 0.03$ | $10.57 \pm 0.00$ |
| | | | penalize | $\mathbf{54.20 \pm 0.02}$ | $\mathbf{46371.00 \pm 0.00}$ | $\mathbf{55.45 \pm 0.03}$ | $\mathbf{10.53 \pm 0.00}$ |
| | 30 | 69557 | simple | $64.91 \pm 0.11\ (50.93 \pm 0.00)$ | $38250.20 \pm 23.68$ | $66.15 \pm 0.11$ | $11.50 \pm 0.01$ |
| | | | top-k | $50.71 \pm 0.02\ (50.93 \pm 0.00)$ | $69557.00 \pm 0.00$ | $51.91 \pm 0.02$ | $10.19 \pm 0.00$ |
| | | | penalize | $\mathbf{50.52 \pm 0.01}$ | $\mathbf{69557.00 \pm 0.00}$ | $\mathbf{51.69 \pm 0.01}$ | $\mathbf{10.17 \pm 0.00}$ |
| *blog* | 10 | 5240 | simple | $2890.77 \pm 240.03\ (297.17 \pm 0.10)$ | $515.00 \pm 5.52$ | $3048.34 \pm 276.60$ | $78.02 \pm 3.47$ |
| | | | top-k | $373.36 \pm 5.20\ (297.17 \pm 0.10)$ | $5240.00 \pm 0.00$ | $287.98 \pm 4.36$ | $24.00 \pm 0.18$ |
| | | | penalize | $\mathbf{326.64 \pm 23.61}$ | $\mathbf{5240.00 \pm 0.00}$ | $\mathbf{217.87 \pm 1.74}$ | $\mathbf{20.87 \pm 0.08}$ |
| | 20 | 10479 | simple | $2209.31 \pm 35.05\ (292.37 \pm 0.01)$ | $712.80 \pm 4.02$ | $2283.30 \pm 41.51$ | $67.57 \pm 0.61$ |
| | | | top-k | $306.08 \pm 1.76\ (292.37 \pm 0.01)$ | $10479.00 \pm 0.00$ | $219.44 \pm 1.76$ | $20.95 \pm 0.08$ |
| | | | penalize | $\mathbf{296.77 \pm 0.34}$ | $\mathbf{10479.00 \pm 0.00}$ | $\mathbf{207.90 \pm 0.43}$ | $\mathbf{20.39 \pm 0.02}$ |
| | 30 | 15719 | simple | $13972.08 \pm 11310.87\ (327.78 \pm 29.11)$ | $10447.00 \pm 23265.78$ | $15134.38 \pm 10849.34$ | $156.81 \pm 84.26$ |
| | | | top-k | $360.31 \pm 27.59\ (327.78 \pm 29.11)$ | $15719.00 \pm 0.00$ | $297.70 \pm 80.29$ | $24.23 \pm 3.24$ |
| | | | penalize | $\mathbf{352.68 \pm 2.16}$ | $\mathbf{15719.00 \pm 0.00}$ | $\mathbf{247.87 \pm 1.11}$ | $\mathbf{22.27 \pm 0.05}$ |

evaluate both methods by optimizing the same upper level function on a test set using the selected $D_{\text{syn}}^{\text{train}}$ sets, reporting the objective value and the root mean square error (RMSE), a common metric for assessing regression model performance. Similar to the group lasso structures discussed in Section 5.1, the *relaxation and rounding*, with both *simple* and *top-k hard thresholding* rounding methodologies, initially achieves a low objective function value at the upper level, but this value increases after rounding. In contrast, the *relax and penalize* method successfully finds an integer solution with a low objective function value. Additionally, the penalty method selects a number of training samples close to the budget, while the *relaxation and rounding* strategy results in a relaxed solution with only a few components above 0.5. The *simple* rounding hence leads to fewer samples and ultimately not utilizing the full budget, while the rounding based on *top-k hard thresholding* naturally leads to a solution that satisfies the budget constraint. When applying the selected $D_{\text{syn}}^{\text{train}}$ to a test set, the *relax and penalize* method always achieves lower values for the objective function and RMSE, leading to a better overall solution with respect to *relaxation and rounding*, both *simple* and *top-k*. In Section D.4, we report the evolution along the different steps of the quantities reported in Table 3 for one setting.

## 6 Conclusion

In this paper, we studied the idea of relaxing the integrality constraints and using a penalty term to handle mixed-binary bilevel optimization problems arising in hyperparameter tuning of machine learning systems. Besides a result concerning the equivalence in terms of global minimizers, sufficient conditions for identifying mixed-binary local minimizers are stated. These theoretical results naturally lead to devise a penalty method that is, under suitable assumptions, guaranteed to provide mixed-binary solutions. The reported numerical results highlight that our method is competitive with state-of-the-art approaches based on relaxation and rounding for the group lasso problem and outperforms these methods on the data distillation task.

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

# A    Auxiliary results

In this section, we give a number of technical results related to the penalty function $\varphi$ used throughout the paper. We recall that $\varphi\colon [0,1]^p \to \mathbb{R}$ is defined as

$$\varphi(\theta) = \sum_{i=1}^{p} \theta_i(1 - \theta_i).$$

Moreover, we denote by $[p]$ the set $\{1, \dots, p\}$ and by $\|\cdot\|$ the Euclidean norm in $\mathbb{R}^p$. Occasionally, we will also used the norms

$$\|\theta\|_1 = \sum_{i=1}^{p} |\theta_i| \quad \text{and} \quad \|\theta\|_\infty = \max_{1 \le i \le p} |\theta_i|.$$

**Lemma 1.** *Let $\psi\colon [0,1] \to \mathbb{R}$ be such that*

$$\forall\, t \in [0,1]\colon \quad \psi(t) = t(1 - t).$$

*Let $\sigma \in\ ]0, 1/2]$. Then, for every $t_1, t_2 \in [0,1]$ we have*

$$\left| \frac{t_1 + t_2}{2} - \frac{1}{2} \right| \ge \sigma \implies |\psi(t_2) - \psi(t_1)| \ge 2\sigma|t_2 - t_1|.$$

*Proof.* Let $t_1, t_2 \in [0, 1]$. One can easily check that

$$\psi(t_2) - \psi(t_1) = (1 - (t_1 + t_2))(t_2 - t_1).$$

Therefore,

$$|\psi(t_2) - \psi(t_1)| = |t_1 + t_2 - 1||t_2 - t_1| = 2\left|\frac{t_1 + t_2}{2} - \frac{1}{2}\right||t_2 - t_1| \geq 2\sigma|t_2 - t_1|. \qquad \square$$

**Lemma 2.** *Let $\sigma \in ]0, 1/2]$ and $\theta, \theta' \in [0, 1]^p$ be such that the following holds:*

$$\forall i \in [p]: \ \theta'_i \neq \theta_i \implies \left|\frac{\theta_i + \theta'_i}{2} - \frac{1}{2}\right| \geq \sigma \quad and \quad \left|\theta'_i - \frac{1}{2}\right| \leq \left|\theta_i - \frac{1}{2}\right|.$$

*Then, $\varphi(\theta') - \varphi(\theta) \geq 2\sigma\|\theta' - \theta\|$.*

*Proof.* Let $\theta, \theta' \in [0, 1]^p$ be as in the statement and let $I = \{i \in [p] \,|\, \theta_i \neq \theta'_i\}$. Then,

$$\forall i \in I: \ \left|\frac{\theta_i + \theta'_i}{2} - \frac{1}{2}\right| \geq \sigma \quad and \quad \psi(\theta_i) \leq \psi(\theta'_i)$$

and hence, by Lemma 1, we have

$$\begin{aligned}
\varphi(\theta') - \varphi(\theta) &= \sum_{i \in I} \psi(\theta'_i) - \psi(\theta_i) \\
&= \sum_{i \in I} |\psi(\theta'_i) - \psi(\theta_i)| \geq 2\sigma \sum_{i \in I} |\theta'_i - \theta_i| \\
&= 2\sigma\|\theta' - \theta\|_1 \geq 2\sigma\|\theta' - \theta\|,
\end{aligned}$$

where we used $\|\cdot\| \leq \|\cdot\|_1$ for the last inequality. $\qquad \square$

**Remark 5.** *Lemma 2 says that if the components of the mid point $(\theta + \theta')/2$ are bounded away from $1/2$ and the componentwise distance from $\theta$ to $1/2$ is larger than that of $\theta'$ to $1/2$, then $\varphi(\theta') - \varphi(\theta)$ can be bounded from below by $\|\theta' - \theta\|$; up to a multiplicative constant.*

**Remark 6.** *The conditions on $\theta$ and $\theta'$ required by Lemma 2 are satisfied if*

$$\forall i \in [p]: \ \theta_i \neq \theta'_i \implies \begin{cases} (\theta_i - 1/2)(\theta'_i - 1/2) \geq 0, \\ |\theta'_i - 1/2| \leq |\theta_i - 1/2|, \\ 2\sigma \leq |\theta_i - 1/2|. \end{cases}$$

*Indeed, in such case we have*

$$\begin{aligned}
\left|\frac{\theta_i + \theta'_i}{2} - \frac{1}{2}\right| &= \frac{1}{2}|\theta_i + \theta'_i - 1| \\
&= \frac{1}{2}\left|\theta_i - \frac{1}{2} + \theta'_i - \frac{1}{2}\right| \stackrel{(*)}{=} \frac{1}{2}\left(\left|\theta_i - \frac{1}{2}\right| + \left|\theta'_i - \frac{1}{2}\right|\right) \geq \frac{1}{2}\left|\theta_i - \frac{1}{2}\right| \geq \sigma,
\end{aligned}$$

*where the equality in $(*)$ is due to the fact that*

$$(\theta_i - 1/2)(\theta'_i - 1/2) \geq 0, \ |\theta'_i - 1/2| \leq |\theta_i - 1/2| \implies (\theta_i \leq \theta'_i \leq 1/2 \ \ or \ \ \theta_i \geq \theta'_i \geq 1/2).$$

**Corollary 1.** *Let $\sigma \in ]0, 1/2]$ and $\theta \in \{0, 1\}^p$. Then*

$$\forall \theta' \in [0, 1]^p: \ \|\theta' - \theta\| \leq 1 - 2\sigma \implies \varphi(\theta') \geq 2\sigma\|\theta' - \theta\|.$$

*Proof.* Let $\theta \in \{0,1\}^p$ and $\theta' \in [0,1]^p$. We will check the conditions in Lemma 2. Let $i \in [p]$. Since $|\theta_i - 1/2| = 1/2$, the condition $|\theta'_i - 1/2| \leq |\theta_i - 1/2|$ is automatically satisfied. Now, we note that

$$\theta_i = 0 \implies \left| \frac{\theta_i + \theta'_i}{2} - \frac{1}{2} \right| = \left| \frac{\theta'_i}{2} - \frac{1}{2} \right| = \frac{1}{2}|\theta'_i - 1| = \frac{1}{2}(1 - |\theta'_i - \theta_i|),$$

$$\theta_i = 1 \implies \left| \frac{\theta_i + \theta'_i}{2} - \frac{1}{2} \right| = \left| \frac{\theta'_i}{2} \right| = \frac{1}{2}|\theta'_i| = \frac{1}{2}(1 - |\theta'_i - \theta_i|).$$

Therefore,

$$\left| \frac{\theta_i + \theta'_i}{2} - \frac{1}{2} \right| \geq \sigma \iff 1 - |\theta'_i - \theta_i| \geq 2\sigma \iff |\theta'_i - \theta_i| \leq 1 - 2\sigma$$

and the first condition in Lemma 2 is then equivalent to the condition $\|\theta' - \theta\|_\infty \leq 1 - 2\sigma$. The statement follows by recalling that $\|\cdot\|_\infty \leq \|\cdot\|$. $\qquad\square$

**Remark 7.** *It is clear from the proof of Corollary 1 that, in fact, it holds*

$$\forall \theta' \in [0,1]^p : \quad \|\theta' - \theta\|_\infty \leq 1 - 2\sigma \implies \varphi(\theta') \geq 2\sigma\|\theta' - \theta\|.$$

**Lemma 3.** *Let $\theta \in \{0,1\}^p$, $\bar{\theta} \in [0,1]^p$, and $c \in \mathbb{R}$ be such that $\|\theta - \bar{\theta}\|_\infty < c < \frac{1}{2}$. Let*

$$\theta^t = (1-t)\bar{\theta} + t\theta \quad \text{with} \quad t \in [0,1].$$

*Then,*

$$\varphi(\bar{\theta}) - \varphi(\theta^t) \geq (1-2c)\|\theta^t - \bar{\theta}\|.$$

*Proof.* Since $|\theta_i - \bar{\theta}_i| < c < \frac{1}{2}$ holds for all $i = 1, \dots, n$, we have

$$\frac{1}{2} = \left| \theta_i - \frac{1}{2} \right| \leq |\theta_i - \bar{\theta}_i| + \left| \bar{\theta}_i - \frac{1}{2} \right| < c + \left| \bar{\theta}_i - \frac{1}{2} \right|,$$

which implies

$$\left| \bar{\theta}_i - \frac{1}{2} \right| > \frac{1}{2} - c.$$

Moreover, since $\theta_i \in \{0,1\}$ and $|\bar{\theta}_i - \theta_i| < c < \frac{1}{2}$, we have

$$\theta_i = 0 \implies \theta_i = 0 \leq \bar{\theta}_i < c < \frac{1}{2},$$

$$\theta_i = 1 \implies \frac{1}{2} < 1 - c < \bar{\theta}_i \leq 1 = \theta_i,$$

and, hence, since $\theta_i^t$ is between $\theta_i$ and $\bar{\theta}_i$, it holds

$$\left| \theta_i^t - \frac{1}{2} \right| \geq \left| \bar{\theta}_i - \frac{1}{2} \right| > \frac{1}{2} - c$$

so that

$$\left| \frac{\theta_i^t + \bar{\theta}_i}{2} - \frac{1}{2} \right| = \left| \frac{1}{2}(\theta_i^t + \bar{\theta}_i - 1) \right| = \frac{1}{2} \left| \left( \theta_i^t - \frac{1}{2} \right) + \left( \bar{\theta}_i - \frac{1}{2} \right) \right|$$

$$= \frac{1}{2} \left( \left| \theta_i^t - \frac{1}{2} \right| + \left| \bar{\theta}_i - \frac{1}{2} \right| \right) \geq \left| \bar{\theta}_i - \frac{1}{2} \right| > \frac{1}{2} - c.$$

Therefore, by Lemma 2, $\varphi(\bar{\theta}) - \varphi(\theta^t) \geq (1-2c)\|\theta^t - \bar{\theta}\|$ holds. $\qquad\square$

## B Proof of Theorem 1

For the sake of brevity, we set

$$S = \underset{(\theta,\lambda)\in\Lambda\times\Theta_{\mathrm{bin}}}{\operatorname{argmin}} \; G(\theta,\lambda) \quad \text{and} \quad S(\varepsilon) = \underset{(\theta,\lambda)\in\Lambda\times\Theta}{\operatorname{argmin}} \; G(\theta,\lambda) + \frac{1}{\varepsilon}\varphi(\theta).$$

Recall that $\Theta_{\mathrm{bin}} = \Theta \cap \{0,1\}^p$. Let $\rho \in ]0,1[$ and let $\hat{\varepsilon} \in ]0,(1-\rho)/L[$. We define the open set

$$U = \bigcup_{\theta\in\Theta_{\mathrm{bin}}} B_\rho(\theta),$$

where $\rho$ is chosen small enough so to ensure that $\Theta \setminus U \neq \varnothing$.[3] Let $\bar{\theta}$ be a minimizer of $\varphi$ over the compact set $\Theta \setminus U$. Then, clearly

$$\forall \theta' \in \Theta \setminus U : \quad \varphi(\theta') \geq \varphi(\bar{\theta}) > 0. \tag{17}$$

(Note that, since $\bar{\theta} \notin U$, then $\bar{\theta} \notin \{0,1\}^p$, and hence there exists $i \in [p]$ such that $\bar{\theta}_i \in ]0,1[$, which implies that $\varphi(\bar{\theta}) \geq \bar{\theta}_i(1-\bar{\theta}_i) > 0$.) Thus, since

$$\lim_{\varepsilon\to 0^+} \frac{1}{\varepsilon}\varphi(\bar{\theta}) = +\infty,$$

there exists $\tilde{\varepsilon} \in ]0,\hat{\varepsilon}]$ such that

$$\forall \varepsilon \in ]0,\tilde{\varepsilon}] : \quad \frac{1}{\varepsilon}\varphi(\bar{\theta}) > \sup_{\Lambda\times\Theta_{\mathrm{bin}}} G - \inf_{\Lambda\times(\Theta\setminus U)} G. \tag{18}$$

Now, we let $\varepsilon \in ]0,\tilde{\varepsilon}]$ and show that $S(\varepsilon) \subset \Theta_{\mathrm{bin}}$. Let $(\lambda^*,\theta^*) \in S(\varepsilon)$ and suppose, by contradiction, that $(\lambda^*,\theta^*) \notin \Theta_{\mathrm{bin}}$. We can have the following two cases:

(a) Let $\theta^* \in U$. Then, there exists $\theta \in \Theta_{\mathrm{bin}}$ such that $\theta^* \in \Theta \cap B_\rho(\theta)$. Thus, in view of Assumption 1 and Corollary 1, we have

$$G(\lambda^*,\theta) - G(\lambda^*,\theta^*) \leq \; L\|\theta^*-\theta\| < \frac{1-\rho}{\varepsilon}\|\theta^*-\theta\| \leq \frac{1}{\varepsilon}\varphi(\theta^*) = \frac{1}{\varepsilon}\varphi(\theta^*) - \frac{1}{\varepsilon}\varphi(\theta)$$

and, hence,

$$G(\lambda^*,\theta) + \frac{1}{\varepsilon}\varphi(\theta) < G(\lambda^*,\theta^*) + \frac{1}{\varepsilon}\varphi(\theta^*),$$

which yields a contradiction since $(\lambda^*,\theta^*)$ is a global minimizer of Problem (3).

(b) Let $\theta^* \notin U$. Then $(\lambda^*,\theta^*) \in \Lambda \times (\Theta \setminus U)$ and hence, recalling (17) and (18), we have

$$\begin{aligned}
G(\lambda^*,\theta^*) + \frac{1}{\varepsilon}\varphi(\theta^*) &\geq \inf_{\Lambda\times(\Theta\setminus U)} G + \frac{1}{\varepsilon}\varphi(\theta^*) \\
&\geq \inf_{\Lambda\times(\Theta\setminus U)} G + \frac{1}{\varepsilon}\varphi(\bar{\theta}) \\
&> \sup_{\Lambda\times\Theta_{\mathrm{bin}}} G \\
&\geq G(\theta,\lambda) + \frac{1}{\varepsilon}\underbrace{\varphi(\theta)}_{=0}
\end{aligned}$$

for any $(\theta,\lambda) \in \Lambda \times \Theta_{\mathrm{bin}} \subseteq \Lambda \times \Theta$, which gives again a contradiction.

---

[3]This means that there exists a $\theta^* \in \Theta$ such that for every $\theta \in \Theta_{\mathrm{bin}}$, it holds $\|\theta - \theta^*\| > \rho$. This condition is met if we pick $\theta^* \in \Theta \setminus \{0,1\}^p$ (see Assumption 1(ii)) and (taking into account that $\Theta_{\mathrm{bin}} = \Theta \cap \{0,1\}^p$ is a finite set) choose $\rho$ such that $0 < \rho < \inf_{\theta\in\Theta_{\mathrm{bin}}} \|\theta - \theta^*\|$.

Thus, in both cases we get a contradiction and therefore necessarily $(\lambda^*, \theta^*) \in \Lambda \times \Theta_{\text{bin}}$. Now, if we take $(\lambda^*, \theta^*) \in S(\varepsilon)$, since $(\lambda^*, \theta^*) \in \Lambda \times \Theta_{\text{bin}}$, we have

$$G(\lambda^*, \theta^*) + \frac{1}{\varepsilon} \underbrace{\varphi(\theta^*)}_{=0} \leq G(\theta, \lambda) + \frac{1}{\varepsilon} \underbrace{\varphi(\theta)}_{=0} \quad \forall (\theta, \lambda) \in \Lambda \times \Theta_{\text{bin}} \subseteq \Lambda \times \Theta.$$

Thus, for all $(\theta, \lambda) \in \Lambda \times \Theta_{\text{bin}}$, $G(\lambda^*, \theta^*) \leq G(\theta, \lambda)$, meaning $(\lambda^*, \theta^*) \in S$. Vice versa, let $(\lambda^*, \theta^*) \in S$. Choosing $(\tilde{\lambda}^*, \tilde{\theta}^*) \in S(\varepsilon)$, since $(\tilde{\lambda}^*, \tilde{\theta}^*) \in \Lambda \times \Theta_{\text{bin}}$, we have

$$G(\lambda^*, \theta^*) + \frac{1}{\varepsilon} \underbrace{\varphi(\theta^*)}_{=0} = G(\lambda^*, \theta^*) \leq G(\tilde{\lambda}^*, \tilde{\theta}^*)$$

$$\leq G(\tilde{\lambda}^*, \tilde{\theta}^*) + \frac{1}{\varepsilon} \varphi(\tilde{\theta}^*) = \min_{(\theta, \lambda)) \in \Lambda \times \Theta} G(\theta, \lambda) + \frac{1}{\varepsilon} \varphi(\theta).$$

Hence, $(\lambda^*, \theta^*) \in S(\varepsilon)$.

## C   Details on the group lasso application

In this section, we report on the details regarding the group-sparsity structure estimation in regression problems presented in Section 5.1. Firstly, we discuss the extension of the algorithm presented in Frecon et al. (2018) to the mix-integer case. Secondly, we report the details of the performed experiments.

### C.1   Extensions to (Frecon et al., 2018)

In Frecon et al. (2018), Problem (9) is considered only in the integer hyperparameter $\theta$, while the real hyperparameter $\lambda$ is supposed to be fixed. Here, we report the extension to the optimization in both the hyperparameters $\theta$ and $\lambda$. In particular, Frecon et al. (2018) do not solve the lower-level problem exactly, rather consider the following approximate problem, providing conditions under which it converges to the exact one as the number of inner iterations $q$ grows:

$$\min_{(\theta, \lambda) \in \Lambda \times \Theta} \mathcal{U}^{(q)}(\theta, \lambda) \quad \text{with} \quad \begin{cases} u^{(0)} \equiv 0 \in \mathbb{R}^{P \times L}, \\ \forall i = 0, 1, \ldots, q-1: \ u^{(i+1)}(\theta, \lambda) = \mathcal{A}(u^{(i)}(\theta, \lambda), \theta, \lambda), \\ w^{(q)}(\theta, \lambda) = \mathcal{B}(u^{(q)}(\theta, \lambda), \theta, \lambda), \\ \mathcal{U}^{(q)}(\theta, \lambda) = \frac{1}{T} \sum_{t=1}^{T} C_t(w^{(q)}(\theta, \lambda)) \end{cases}$$

and $\mathcal{A} : \mathbb{R}^{P \times L} \times \Lambda \times \Theta \to \mathbb{R}^{P \times L}$ as well as $\mathcal{B} : \mathbb{R}^{P \times L} \times \Lambda \times \Theta \to \mathbb{R}^P$. We denote by $\partial_1 \mathcal{A}(u, \theta, \lambda)$ the partial derivatives of $\mathcal{A}$ with respect the variable $u$ and $\partial_2 \mathcal{A}(u, \theta, \lambda)$ the partial derivatives of $\mathcal{A}$ with respect the variables $\lambda$ and $\theta$. The same notation is used for the partial derivatives of $\mathcal{B}$. When specializing to the case of group lasso, $\mathcal{A}$ and $\mathcal{B}$ take the expression reported in Section B.1 of Frecon et al. (2018) supplementary material:

$$\mathcal{A}(u, \theta, \lambda) = \nabla \Phi^*_\lambda(\nabla \Phi_\lambda(u) + \gamma A_\theta \mathcal{B}(u, \theta, \lambda)),$$
$$\mathcal{B}(u, \theta) = \nabla f^*(-A_\theta^\top u)$$

with $\Phi^*_\lambda$ being the separable Hellinger-like function as defined in Definition 3.2 in Frecon et al. (2018), $\gamma > 0$ is some given step-size, $f^*$ is the Fenchel conjugate of $f$, and $A_\theta^\top$ is the transpose of the operator $A_\theta$ as defined in Problem 3.1 and Problem 3.2 in Frecon et al. (2018). Therefore, noticing that the dependence on $\theta$ is hidden in $\Phi_\lambda$, we only need to update $\partial_2 \mathcal{A}(u, \theta, \lambda)$, because $\mathcal{B}$ does not depend on $\theta$ and $\partial_1 \mathcal{A}(u, \theta, \lambda)$ is the derivative by $u$. We recall that, for every $u = (u_l)_{1 \geq l \leq L} \in \mathbb{R}^{P \times L}$ and $v = (v_l)_{1 \geq l \leq L} \in \mathbb{R}^{P \times L}$ for every $l = 1, \ldots, L$,

$$\nabla_l \Phi(u) = \nabla \phi(u_l) = \frac{u_l}{\sqrt{\lambda^2 - \|u_l\|_2^2}} \quad \text{and} \quad \nabla_l \Phi^*(v) = \nabla \phi^*(v_l) = \frac{v_l}{\sqrt{\lambda^2 - \|v_l\|_2^2}}$$

holds. Therefore, we obtain

$$\mathcal{A}^{(l)}(u, \theta, \lambda) = \lambda \left( \frac{v_l(\lambda)}{\sqrt{1 + \|v_l(\lambda)\|_2^2}} \right)$$

and

$$\partial_\lambda \mathcal{A}^{(l)}(u, \theta, \lambda) = \frac{v_l(\lambda) + \lambda v_l'(\lambda)}{\sqrt{1 + \|v_l(\lambda)\|_2^2}} - \frac{\lambda v_l(\lambda) \langle v_l(\lambda), v_l'(\lambda) \rangle}{(1 + \|v_l(\lambda)\|_2^2)^{\frac{3}{2}}}$$

with

$$v_l(\lambda) = \frac{1}{\sqrt{1 + \|u_l\|_2^2}} \cdot u_l + \gamma \theta_l \odot \mathcal{B}(u, \theta) \quad \text{and} \quad v_l'(\lambda) = -\frac{\lambda}{(1 + \|u_l\|_2^2)^{\frac{3}{2}}} \cdot u_l.$$

For clarity, we re-write $v_l(\lambda)$ and $v_l'(\lambda)$ as

$$v_l(\lambda) = \iota \cdot u_l + d_l, \quad v_l'(\lambda) = \kappa \cdot u_l$$

with

$$\iota = \frac{1}{\sqrt{1 + \|u_l\|_2^2}}, \quad \kappa = \frac{-\lambda}{(1 + \|u_l\|_2^2)^{\frac{3}{2}}}, \quad d_l = \gamma \theta_l \odot \mathcal{B}(u, \theta).$$

Our aim is to compute $\partial_2 \mathcal{A}(u, \theta, \lambda)^\top a$ with $a \in \mathbb{R}^{P \times L}$, where the partial derivative is with respect to all the hyperparameters, meaning the group of variables $(\theta, \lambda)$. Considering that

$$\forall (b, \beta) \in \mathbb{R}^{P \times L} \times \mathbb{R}: \quad \partial_2 \mathcal{A}(u, \theta, \lambda)(b, \beta) = \partial_\theta \mathcal{A}(u, \theta, \lambda)b + \partial_\lambda \mathcal{A}(u, \theta, \lambda)\beta,$$

we have

$$\begin{aligned}
\langle \partial_2 \mathcal{A}(u, \theta, \lambda)(b, \beta), a \rangle &= \langle \partial_\theta \mathcal{A}(u, \theta, \lambda)b, a \rangle + \beta \langle \partial_\lambda \mathcal{A}(u, \theta, \lambda), a \rangle \\
&= \langle b, \partial_\theta \mathcal{A}(u, \theta, \lambda)^\top a \rangle + \langle \partial_\lambda \mathcal{A}(u, \theta, \lambda), a \rangle \cdot \beta \\
&= \langle (b, \beta), (\partial_\theta \mathcal{A}(u, \theta, \lambda)^\top a, \langle \partial_\lambda \mathcal{A}(u, \theta, \lambda), a \rangle) \rangle.
\end{aligned} \tag{19}$$

It follows that

$$\partial_2 \mathcal{A}(u, \theta, \lambda)^\top a = \left( \partial_\theta \mathcal{A}(u, \theta, \lambda)^\top a, \langle \partial_\lambda \mathcal{A}(u, \theta, \lambda), a \rangle \right).$$

Thanks to this result, we can use Algorithm 2 in Frecon et al. (2018) extended to the optimization in both hyperparameters to the hypergradient computation, substituting the expression of $\partial_2 \mathcal{A}(u^{(i)}(\theta, \lambda), \theta, \lambda)^\top a^{(i+1)}$ in the calculation of $b^{(i)}$ (with $i$ iteration index); see Algorithm 2. In particular, in our case, we have an additional component of $b^{(i)}$ regarding the hyperparameter $\lambda$, that we call $\beta^{(i)} \in \mathbb{R}$. Therefore,

$$\begin{aligned}
\beta^{(i)} &= \langle \partial_\lambda \mathcal{A}(u^{(i)}(\theta, \lambda), \theta, \lambda), a^{(i+1)} \rangle + \beta^{(i+1)} \\
&= \sum_{l=1}^{L} \langle \partial_\lambda \mathcal{A}^{(l)}(u^{(i)}(\theta, \lambda), \theta, \lambda), a_l^{(i+1)} \rangle + \beta^{(i+1)}
\end{aligned}$$

holds with

$$\langle \partial_\lambda \mathcal{A}^{(l)}(u^{(i)}(\theta, \lambda), \theta, \lambda), a_l^{(i+1)} \rangle = \frac{\langle v_l(\lambda) + \lambda v_l'(\lambda), a_l^{(i+1)} \rangle}{\sqrt{1 + \|v_l(\lambda)\|^2}} - \frac{\lambda \langle v_l(\lambda), v_l'(\lambda) \rangle \langle v_l(\lambda), a_l^{(i+1)} \rangle}{(1 + \|v_l(\lambda)\|)^{\frac{3}{2}}}.$$

### C.2 Experimental setup

In Section 5.1, we conduct experiments on synthetic datasets for the group lasso problem. Regarding the details of the experiments, we implement the framework of Algorithm 1 by solving a sequence of outer optimization problems $P^k$ of type (10) w.r.t. $(\lambda, \theta)$. We select $\varepsilon^0 = 10^5$, such that the penalty term does not dominate the function, making the evaluation at the first point behave similarly to the unconstrained version. In order to better exploit the hyperparameter $\beta$, we perform an ablation study for the setting with inequal group sizes and $a = 0.3$. From Table 4, we can notice that a high $\beta$ leads to higher number of external

---

**Algorithm 2:** Hypergradient computation (reverse mode)

---

1 **Require:** Group structure $\theta$, number of inner iterations $q$.

2     Initialize $u^{(0)}(\theta, \lambda) \equiv 0 \in \mathbb{R}^{P \times L}$

3     **for** $i = 1$ to $q$ **do**

4         $u^{(i)}(\theta, \lambda) = \mathcal{A}(u^{(i-1)}(\theta, \lambda), \theta, \lambda).$

5     **end for**

6 **Output:** 1. $u^{(0)}(\theta, \lambda), \ldots, u^{(q)}(\theta, \lambda),\ w^{(q)}(\theta, \lambda) = \mathcal{B}(u^{(q)}(\theta, \lambda), \theta).$

7     Initialize $a_q = \partial_1 \mathcal{B}(u^{(q)}(\theta, \lambda), \theta, \lambda)^\top \nabla C(x^{(q)}(\theta, \lambda)), b_q = \partial_2 \mathcal{B}(u^{(q)}(\theta, \lambda), \theta, \lambda)^\top \nabla C(x^{(q)}(\theta, \lambda)).$

8     **for** $i = q - 1$ to $0$ **do**

9         $a^{(i)} = \partial_1 \mathcal{A}(u^{(i)}(\theta, \lambda), \theta, \lambda)^\top a^{(i+1)}$

10       $b^{(i)} = \partial_\theta \mathcal{A}(u^{(i)}(\theta, \lambda), \theta, \lambda)^\top a^{(i+1)} + b^{(i+1)}$

11       $\beta^{(i)} = \langle \partial_\lambda \mathcal{A}(u^{(i)}(\theta, \lambda), \theta, \lambda), a^{(i+1)} \rangle + \beta^{(i+1)}$

12     **end for**

13 **Output:** 2. Hypergradients $\nabla_\theta \mathcal{U}^{(q)}(\theta, \lambda) = b^{(0)}$, $\nabla_\lambda \mathcal{U}^{(q)}(\theta, \lambda) = \beta^{(0)}$.

---

Table 4: Ablation study hyperparameter $\beta$ for the setting inequal group size and $a = 0.3$.

| $\beta$ | outer iter | $G(\lambda^p, \theta^p)$ | $\|w(\lambda^p, \theta^p) - w^\star\|_F$ |
|---|---|---|---|
| 0.1 | $4.33 \pm 0.58$ | $0.06 \pm 0.00$ | $5.86 \pm 0.17$ |
| 0.3 | $7.00 \pm 0.00$ | $0.05 \pm 0.00$ | $5.71 \pm 0.14$ |
| 0.5 | $\mathbf{11.33 \pm 0.58}$ | $\mathbf{0.05 \pm 0.00}$ | $\mathbf{5.54 \pm 0.11}$ |
| 0.7 | $18.00 \pm 0.00$ | $0.06 \pm 0.03$ | $5.73 \pm 0.68$ |
| 0.9 | $47.00 \pm 5.29$ | $0.07 \pm 0.03$ | $5.87 \pm 1.02$ |

iterations and a longer running time. Across the different $\beta$, the algorithm seems to find different local minima. The best performances, in terms of test error and reconstruction errors, are reached with $\beta = 0.5$, which we used for all the experiments. The stopping criterion is the condition $\mathrm{dist}_\infty(\theta^k, \{0, 1\}^{P \times L}) < tol$ with $tol = 10^{-2}$ tolerance. The lower-level problem in (8) is solved using Algorithm 1 described in Frecon et al. (2018) stochastically, setting the batch size to 1 as in their paper. Therefore, at each iteration, we consider one $w_t$, $\eta = 10^{-3}$, $q = 500$ inner iterations, and $0.99\,\eta/\lambda$ as the inner step size. For the upper-level optimization in (10), we utilize SAGA (Defazio et al., 2014) and we distribute the inner iterations for each external iteration as follows: $[5000, 5000, 2500, 2500, 2500, 2500, 2500, 1000, 1000, \ldots]$. The step size is set to $T/0.025$ for $\theta$ and it is multiplied by the preconditioner $c = 10^{-4}$ for $\lambda$. The hyperparameters $\theta$ and $\lambda$ are projected to the unit simplex $(\Delta^{L-1})^P$ and the box $[10^{-3}, 1]$, respectively, and they are initialized to $\lambda^0 = 10^{-1}$ and $\theta^0 = \mathcal{P}_\Theta(L^{-1}\mathrm{I}_{P \times L} + \mathcal{N}(0_{P \times L}, 0.1 L^{-1}\mathrm{I}_{P \times L}))$. For a fair comparison, in Figure 1 and Tables 1 and 2 we run all the methods with the same parameters and the same amount of total iterations. Here we report also in Figure 2 the evolution of the quantity $\mathrm{dist}_\infty(\theta^k, \Theta_{\mathrm{bin}})$ along the iterations for the *relax and penalize* method, showing that for $k$ sufficiently large it abruptly reaches 0 as soon as it is below $\frac{1}{2}$ as predicted in Remark 3.

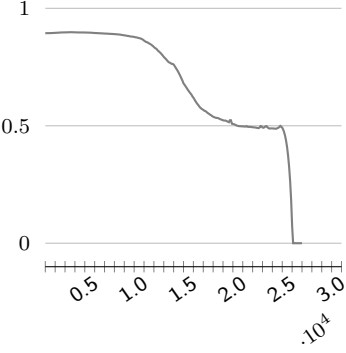

Figure 2: Evolution of $\mathrm{dist}_\infty(\theta^k, \Theta_{\mathrm{bin}})$ for the setting with inequal group sizes and $a = 0.3$.

# D   Details on the data distillation application

In this section, we report the calculations regarding the data distillation issue presented in Section 5.2. Firstly, we present the calculations to solve exactly the lower-level problem. Secondly, we report the calculations of the gradient of the upper-level problem needed to perform the stochastic gradient descent. Finally, we report the details of the experiments that were performed.

## D.1   Lower-level calculations

The lower-level problem that we want to solve in (13) is given by

$$\min_{W,b} \quad \frac{1}{m}\sum_{i=1}^{m} v_i\|Wx_i^{\text{train}} + b - y_i^{\text{train}}\|^2 + s\|W\|^2 \tag{20}$$

with $x_i^{\text{train}} \in \mathbb{R}^d$, $y_i^{\text{train}} \in \mathbb{R}^e$, $W \in \mathbb{R}^{e \times d}$, $b \in \mathbb{R}^e$. In the subsequent sections, we will use $x$ and $y$ instead of $x^{\text{train}}$ and $y^{\text{train}}$ for simplicity. We define these quantities

$$\bar{x} = \frac{\sum_{i=1}^{m} v_i x_i}{\sum_{i=1}^{m} v_i}, \quad \bar{y} = \frac{\sum_{i=1}^{m} v_i y_i}{\sum_{i=1}^{m} v_i}, \quad \hat{x}_i = x_i - \bar{x}, \quad \hat{y}_i = y_i - \bar{y}.$$

It follows that

$$\sum_{i=1}^{m} v_i \hat{x}_i = \sum_{i=1}^{m} v_i x_i - \sum_{i=1}^{m} v_i \bar{x}_i = 0, \quad \sum_{i=1}^{m} v_i \hat{y}_i = 0. \tag{21}$$

Considering the first part of the objective function in (20), we obtain

$$\begin{aligned}
\sum_{i=1}^{m} v_i\|Wx_i + b - y_i\|^2 &= \sum_{i=1}^{m} v_i\|W\hat{x}_i - \hat{y}_i + W\bar{x} + b - \bar{y}\|^2 \\
&= \sum_{i=1}^{m} v_i\|W\hat{x}_i - \hat{y}_i\|^2 + \sum_{i=1}^{m} v_i\|W\bar{x} + b - \bar{y}\|^2 \\
&\quad + 2\sum_{i=1}^{m} v_i\langle W\sum_{i=1}^{m} v_i\hat{x}_i - \sum_{i=1}^{m} v_i\hat{y}_i, W\bar{x} + b - \bar{y}\rangle \\
&= \sum_{i=1}^{m} v_i\|W\hat{x}_i - \hat{y}_i\|^2 + \sum_{i=1}^{m} v_i\|W\bar{x} + b - \bar{y}\|^2.
\end{aligned} \tag{22}$$

Using this, Problem (20) is equivalent to

$$\min_{W,b} \quad \frac{1}{m}\sum_{i=1}^{m} v_i\|W\hat{x}_i - \hat{y}_i\|^2 + s\|W\|^2 + \frac{1}{m}\left(\sum_{i=1}^{m} v_i\right)\|W\bar{x} + b - \bar{y}\|^2.$$

Therefore, the minimization can be performed separately in the variables $w$ and $b$, and Problem (20) is equivalent to

$$\min_{W} \quad \frac{1}{m}\sum_{i=1}^{m} v_i\|W\hat{x}_i - \hat{y}_i\|^2 + s\|W\|^2 \quad \text{and} \quad b = \bar{y} - W\bar{x}.$$

We solve the first minimization in $W$ by setting

$$0 = \sum_{i=1}^{m} \frac{2}{m}v_i(W\hat{x}_i - \hat{y}_i)\hat{x}_i^\top + 2sW = 2W\left(\frac{1}{m}\sum_{i=1}^{m} v_i\hat{x}_i\hat{x}_i^\top + s\text{Id}\right) - \frac{2}{m}\sum_{i=1}^{m} v_i\hat{y}_i\hat{x}_i^\top,$$

which is valid if and only if

$$W = \left(\frac{1}{m}\sum_{i=1}^{m} v_i\hat{y}_i\hat{x}_i^\top\right)\left(\frac{1}{m}\hat{x}_i\hat{x}_i^\top + s\text{Id}\right)^{-1}.$$

Finally, we obtain the formula in (14). Computationally, we calculate $W(v)z$ for $z \in \mathbb{R}^d$ by first solving

$$(C_v(X) + s\mathrm{Id})a = z$$

in $a$ and then calculating

$$W(v)z = C_v(X, Y)a.$$

### D.2 Upper-level calculations

We consider the upper-level problem in (13) neglecting the constraints on $v$:

$$\min_{v \in \mathbb{R}^m} \quad \frac{1}{2n} \sum_{j=1}^{n} \|W(v)x_j^{\mathrm{val}} + b(v) - y_j^{\mathrm{val}}\|^2. \tag{23}$$

For simplicity, in the subsequent sections, we will use $x$ and $y$ instead of $x^{\mathrm{val}}$ and $y^{\mathrm{val}}$, and we will refer to the objective function in (23) as $J(v)$ with $J : \mathbb{R}^n \to \mathbb{R}$.

To solve the problem with a stochastic gradient descent algorithm, we need to calculate the gradient of $J(v)$:

$$
\begin{aligned}
\frac{\partial J(v)}{\partial v_j} &= \frac{1}{n} \sum_{j=1}^{n} (W(v)x_i + b(v) - y_i)^\top \left( \frac{\partial W(v)}{\partial v_j} x_i + \frac{\partial b(v)}{\partial v_j} \right) \\
&= \frac{1}{n} \sum_{j=1}^{n} (W(v)(x_i - \bar{x}_v) + \bar{y}_v - y_i)^\top \left( \frac{\partial W(v)}{\partial v_j}(x_i - \bar{x}_v) + \frac{1}{\sum_{i=1}^{m} v_i} \hat{y}_j - W(v)\hat{x}_j \right) \\
&= \frac{1}{n} \sum_{j=1}^{n} (W(v)(x_i - \bar{x}_v) + \bar{y}_v - y_i)^\top \frac{\partial W(v)}{\partial v_j}(x_i - \bar{x}_v) \\
&\quad + \frac{1}{\sum_{i=1}^{m} v_i} (W(v)(x_i - \bar{x}_v) + \bar{y}_v - \bar{y})^\top (\hat{y}_j - W(v)\hat{x}_j).
\end{aligned}
\tag{24}
$$

We will proceed to calculate $\frac{\partial W(v)}{\partial v_j}$, taking advantage of the expression given in (14). We consider the following three maps

$$\phi : \mathbb{R}^n \to \mathbb{R}^{n \times d}, \quad v \mapsto \sum_{i=1}^{n} v_i (y_i - \bar{y}_v)(x_i - \bar{x}_v)^\top,$$

$$\psi : \mathbb{R}^n \to \mathbb{R}^{d \times d}, \quad v \mapsto \sum_{i=1}^{n} v_i (x_i - \bar{x}_v)(x_i - \bar{x}_v)^\top + s\mathrm{Id},$$

$$\varphi : \mathbb{GL}(d) \to \mathbb{R}^{d \times d}, \quad A \mapsto A^{-1}$$

with $\mathbb{GL}(d)$ being the general linear group of degree $d$. Therefore, we can write

$$W(v) = \phi(v)\varphi(\psi(v)) \in \mathbb{R}^{n \times d}. \tag{25}$$

Notice that

$$\varphi'(A) : \mathbb{R}^{d \times d} \to \mathbb{R}^{d \times d}, \quad U \mapsto A^{-1}UA^{-1} \quad \forall U \in \mathbb{R}^{d \times d} \tag{26}$$

since

$$\frac{\varphi(A + tU) - \varphi(A)}{t} = \frac{(A + tU)^{-1} - A^{-1}}{t} = A^{-1} \frac{A - (A + tU)}{t}(A + tU)^{-1} = A^{-1}U(A + tU)^{-1}. \tag{27}$$

Using (25) and (27), we can write

$$
\begin{aligned}
\frac{W(v + tu) - W(v)}{t} &= \frac{\phi(v + tu)\varphi(\psi(v + tu)) - \phi(v)\varphi(\psi(v))}{t} \\
&= \frac{1}{t} (\phi(v + tu)\varphi(\psi(v + tu)) - \phi(v + tu)\varphi(\psi(v)) \\
&\quad + \phi(v + tu)\varphi(\psi(v)) - \phi(v)\varphi(\psi(v))) \\
&= \phi(v + tu)\frac{\varphi(\psi(v + tu)) - \varphi(\psi(v))}{t} + \frac{\phi(v + tu) - \phi(v)}{t}\varphi(\psi(v)).
\end{aligned}
\tag{28}
$$

It follows that

$$
\begin{aligned}
W'(v)[u] &= \phi(v)(\varphi \circ \psi)'(v)[u] + \phi'(v)[u]\varphi(\psi(v)) \\
&= \phi(v)\psi(v)^{-1}\psi'(v)[u]\psi(v)^{-1} + \phi'(v)[u]\psi(v)^{-1} \\
&= (C_v(X,Y)(C_v(X)+s\mathrm{Id})^{-1}\psi'(v)[u] + \phi'(v)[u])(C_v(X)+s\mathrm{Id})^{-1} \in \mathbb{R}^{n \times d}
\end{aligned}
\tag{29}
$$

holds, where, in the second equality we used that

$$
(\varphi \circ \psi)'(v)[u] = \varphi'(\psi(v))(\psi'(v)[u]) = \psi(v)^{-1}\psi'(v)[u]\psi(v)^{-1}.
$$

Choosing $u = e_j$ in (29), we obtain

$$
\frac{\partial W(v)}{\partial v_j} = (C_v(X,Y)(C_v(X)+s\mathrm{Id})^{-1}\frac{\partial \psi(v)}{\partial v_j} + \frac{\partial \phi(v)}{\partial v_j}(C_v(X)+s\mathrm{Id})^{-1}.
\tag{30}
$$

From the definitions of $\bar{x}_v$ and $\bar{y}_v$, we retrieve

$$
\begin{aligned}
\frac{\partial \bar{x}_v}{\partial v_j} &= \frac{\partial}{\partial v_j}\left(\frac{\sum_{i=1}^m v_i x_i}{\sum_{i=1}^m v_i}\right) = -\frac{\sum_{i=1}^m v_i x_i}{\left(\sum_{i=1}^m v_i\right)^2} + \frac{x_j}{\sum_{i=1}^m v_i} = \frac{x_j - \bar{x}_v}{\sum_{i=1}^m v_i} = \frac{\hat{x}_j}{\sum_{i=1}^m v_i}, \\
\frac{\partial \bar{y}_v}{\partial v_j} &= \frac{y_j - \bar{y}_v}{\sum_{i=1}^m v_i} = \frac{\hat{y}_j}{\sum_{i=1}^m v_i}.
\end{aligned}
$$

Therefore,

$$
\begin{aligned}
\frac{\partial \phi(v)}{\partial v_j} &= \sum_{i=1}^n \delta_{ij}(y_i - \bar{y}_v)(x_i - \bar{x}_v)^\top + \sum_{i=1}^n v_i \frac{\partial}{\partial v_j}(\bar{y}_v - y_i)(\bar{x}_v - x_i)^\top \\
&= \hat{y}_j \hat{x}_j^\top - \frac{\hat{y}_j \sum_{i=1}^n v_i \hat{x}_i}{\sum_{i=1}^n v_i} - \frac{\left(\sum_{i=1}^n v_i \hat{y}_i\right)\hat{x}_j^\top}{\sum_{i=1}^n v_i} \\
&= \hat{y}_j \hat{x}_j^\top,
\end{aligned}
$$

where we used (21) in the last equation as well as

$$
\frac{\partial \psi(v)}{\partial v_j} = \hat{x}_j \hat{x}_j^\top.
$$

Finally,

$$
\frac{\partial W(v)}{\partial v_j} = \left(C_v(X,Y)(C_v(X)+s\mathrm{Id})^{-1}\hat{X}_j \hat{X}_j^\top + \hat{Y}_j \hat{X}_j^\top\right)(C_v(X)+s\mathrm{Id})^{-1},
$$

and

$$
\begin{aligned}
\frac{\partial b(v)}{\partial v_j} &= \frac{\partial \bar{y}_v}{\partial v_j} - \frac{\partial W(v)}{\partial v_j}\bar{X}_v - W(v)\frac{\partial \bar{x}_v}{\partial v_j} \\
&= \frac{\hat{y}_j}{\sum_{i=1}^n v_i} - \frac{\partial W(v)}{\partial v_j}\bar{X}_v - W(v)\frac{\hat{x}_j}{\sum_{i=1}^n v_i} \\
&= \frac{\hat{y}_j - W(v)\hat{x}_j}{\sum_{i=1}^n v_i} - \frac{\partial W(v)}{\partial v_j}\hat{x}_v.
\end{aligned}
$$

As for the lower lever, computationally we calculate $\frac{\partial W(v)}{\partial v_j}z$ for $z \in \mathbb{R}^d$, solving the following systems

$$
(C_v(X)+s\mathrm{Id})a_j = \hat{x}_j, \quad (C_v(X)+s\mathrm{Id})a = z.
$$

After solving the upper-level problem with the stochastic gradient descent method, we need to project the solution onto the simplex defined by the knapsack constraint. To do this efficiently, we utilize the Kiwiel algorithm (Kiwiel, 2008).

Table 5: Ablation study hyperparameter $\beta$ for the setting *perc*= 20%.

| | *music* | | | | *blog* | | | |
|---|---|---|---|---|---|---|---|---|
| $\beta$ | outer iter | $\ell^{\mathrm{val}}$ | $\ell^{\mathrm{test}}$ | RMSE | outer iter | $\ell^{\mathrm{val}}$ | $\ell^{\mathrm{test}}$ | RMSE |
| 0.1 | $14.2 \pm 0.45$ | $58.77 \pm 0.12$ | $60.59 \pm 0.12$ | $11.01 \pm 0.01$ | $15.2 \pm 0.45$ | $309.24 \pm 0.53$ | $234.94 \pm 0.78$ | $21.68 \pm 0.04$ |
| 0.3 | $25.8 \pm 0.45$ | $58.16 \pm 0.15$ | $59.63 \pm 0.16$ | $10.92 \pm 0.01$ | $28.0 \pm 0.00$ | $306.75 \pm 0.87$ | $229.68 \pm 2.39$ | $21.43 \pm 0.11$ |
| 0.5 | $43.0 \pm 0.00$ | $57.30 \pm 0.09$ | $58.61 \pm 0.10$ | $10.83 \pm 0.01$ | $47.0 \pm 0.00$ | $305.29 \pm 0.34$ | $225.26 \pm 2.11$ | $21.23 \pm 0.10$ |
| 0.7 | $81.2 \pm 0.45$ | $56.03 \pm 0.07$ | $57.28 \pm 0.08$ | $10.70 \pm 0.01$ | $88.8 \pm 0.84$ | $302.71 \pm 0.83$ | $219.09 \pm 1.81$ | $20.93 \pm 0.09$ |
| **0.9** | $\mathbf{258.0 \pm 1.87}$ | $\mathbf{54.23 \pm 0.03}$ | $\mathbf{55.48 \pm 0.03}$ | $\mathbf{10.53 \pm 0.00}$ | $\mathbf{291.3 \pm 129.40}$ | $\mathbf{296.77 \pm 25.52}$ | $\mathbf{207.90 \pm 18.13}$ | $\mathbf{20.39 \pm 0.85}$ |

## D.3 Experimental setup

In Section 5.2, we present experiments on the data distillation problem for two regression tasks involving the following real-world datasets:

*music* **(Bertin-Mahieux, 2011)** is a dataset that includes song features from 1922 to 2011. It consists of 463 715 training samples, with the first 231 857 used for training the lower level and the remaining 231 857 reserved for testing the weights afterward. Additionally, 51 630 validation samples were utilized for the upper level. Each sample represents a song, featuring 90 attributes (12 related to timbre average and 78 related to timbre covariance), with the year of release as the target variable (as an integer). The aim is to predict the release year of a song based on its audio features.

*blog* **(Buza, 2014)** is a dataset containing features extracted from blog posts. It comprises 52 397 training and 7624 validation samples, with the first 1089 used for training the lower level and the remaining 6535 set aside for testing the weights afterward. Each sample represents a post with 280 features, and the target variable is the number of comments received in the next 24 hours (as an integer). The goal is to predict comments received in the next 24 hours using various features.

Regarding the details of the experiments, we implement the framework of Algorithm 1 by solving a sequence of $K$ outer optimization problems ($\mathrm{P}^k$) of type (16) w.r.t. $v$. The parameters are selected as explained in C.2. In particular, we initialize $\varepsilon^0 = 10^9$ for both datasets, such that the penalty term does not dominate the function, making the evaluation at the first point behave similarly to the unconstrained version. In order to better exploit the hyperparameter $\beta$, we perform an ablation study for the setting *perc*= 20%. From Table 5, we can notice that a high $\beta$ will create more thresholds until convergence, a higher leading to higher number of external iterations and a longer running time. Across the different $\beta$, the algorithm seems to find different local minima. The best performances, in terms of test error and reconstruction errors, are reached with $\beta = 0.9$, which we used for all the experiments. The true stopping criterion is the condition $\mathrm{dist}_\infty(v^k, \{0,1\}^m) < tol$ with $tol = 10^{-2}$ tolerance. For each problem of type (16), we solve the lower-level problem exactly and the upper-level problem with stochastic gradient descent. In the lower-level problem, we set the regularization parameter to $s = 10^2$. For the upper-level problem, we use a batch of size 600 for computing time reasons, we perform 100 inner iterations for each problem ($\mathrm{P}^k$), and we set the step size to $10^{-3}$ for *music* and to $10^{-5}$ for *blog*. The hyperparameter $v$ is projected onto the simplex defined by the knapsack constraint and initialized at $\frac{\tau}{m}\mathbf{I}_m$ with $m$ being the size of the training set and $\tau$ being the budget. For a fair comparison with the *relaxation and rounding*, with both *simple* and *top-k hard thresholding* rounding, we run the code with the same parameters and the same total number of iterations, rounding $v$ at the end. We perform the projection onto the feasible set defined by the knapsack constraints using the Kiwiel algorithm with tolerance $10^{-10}$ and $10^3$ number of iterations.

We report in Figure 3 the evolution of the quantity $\mathrm{dist}_\infty(v^k, \Theta_{\mathrm{bin}})$ along the iterations for the *relax and penalize* method, showing that for $k$ sufficiently large it rapidly reaches 0 as soon as it is below $\frac{1}{2}$ as seen in Figure 2 for the group lasso problem.

## D.4 Additional results along the steps

In Figure 4-6, we report the evolution along the different iterations of the quantities reported in Table 3 for the *music* dataset with distillation budget at 20% of the training set size. Since the solution can also violate integrality constraints through the iterations, the quantities $\left|D_{\mathrm{syn}}^{\mathrm{train}}\right|$, $\ell^{\mathrm{test}}$, and RMSE, are referring to

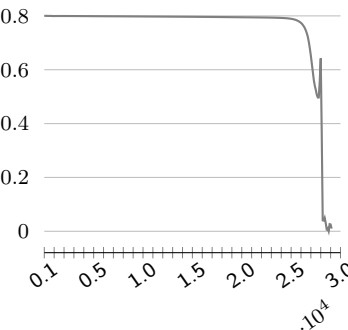

Figure 3: Evolution of $\text{dist}_\infty(v^k, \Theta_{\text{bin}})$ for the *blog* dataset and budget 20%.

the correspondent rounded value. The y-axis represents the internal iterations. Notice that, for example, the violation of the integrality constraints for the *relax and penalize* method is not checked at each internal iteration, but only for the outer ones. We recall that all the methods under analysis evolve with the same total number of iterations and start from the same point. Figure 4 and 5 shows the results regarding the *relaxation and rounding* method, respectively using *simple* and *top-k hard thresholding* rounding. We show the evolution of $\ell^{\text{val}}$ and $\ell^{\text{test}}$ both before and after rounding the solution (resp. in orange and teal), noticing that the objective function values increase after rounding. It can be observed that, for the *relaxation and rounding* method with *simple* rounding, the quantity $\left|D_{\text{syn}}^{\text{train}}\right|$ increases over the iterations but remains well below the total available budget (46371), while the *top-k hard thresholding* use the whole budget at each iteration by definition. For the *relaxation and rounding* method, $\text{dist}_\infty(v, \Theta_{\text{bin}})$ stays stable at 0.5, therefore, no integer solution is found with both the rounding procedures. Figure 5 shows the results regarding the *relax and penalize* method. From the plots of $\ell^{\text{val}}$, $\ell^{\text{test}}$ and RMSE, we can observe an increase after $25,000$ inner iterations, when the penalty term has a higher weight in $(\text{P}^k)$. Corresponding to this, it is apparent that the method finds an integer solution ($\text{dist}_\infty(v, \Theta_{\text{bin}}) = 0$) and that it saturates the budget ($\left|D_{\text{syn}}^{\text{train}}\right| = 46371$).

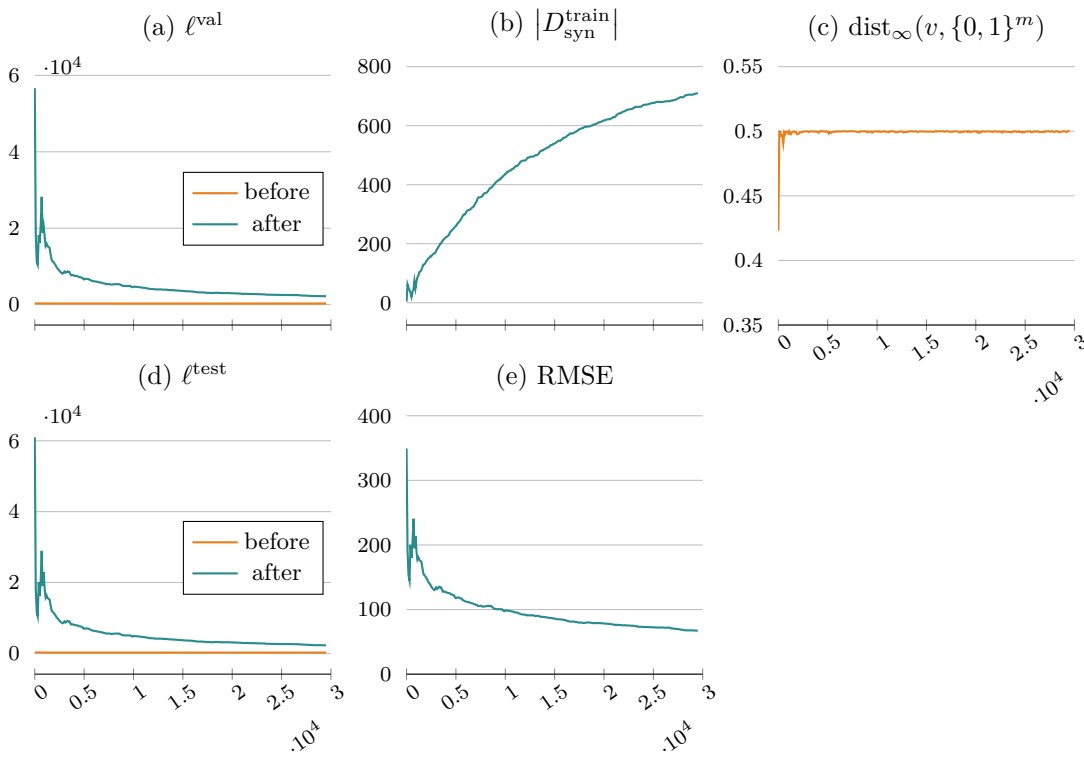

Figure 4: Evolution of the quantities reported in Table 3 for the *relaxation and rounding* method with *simple* rounding on the *music* dataset with distillation budget at 20% of the training set size.

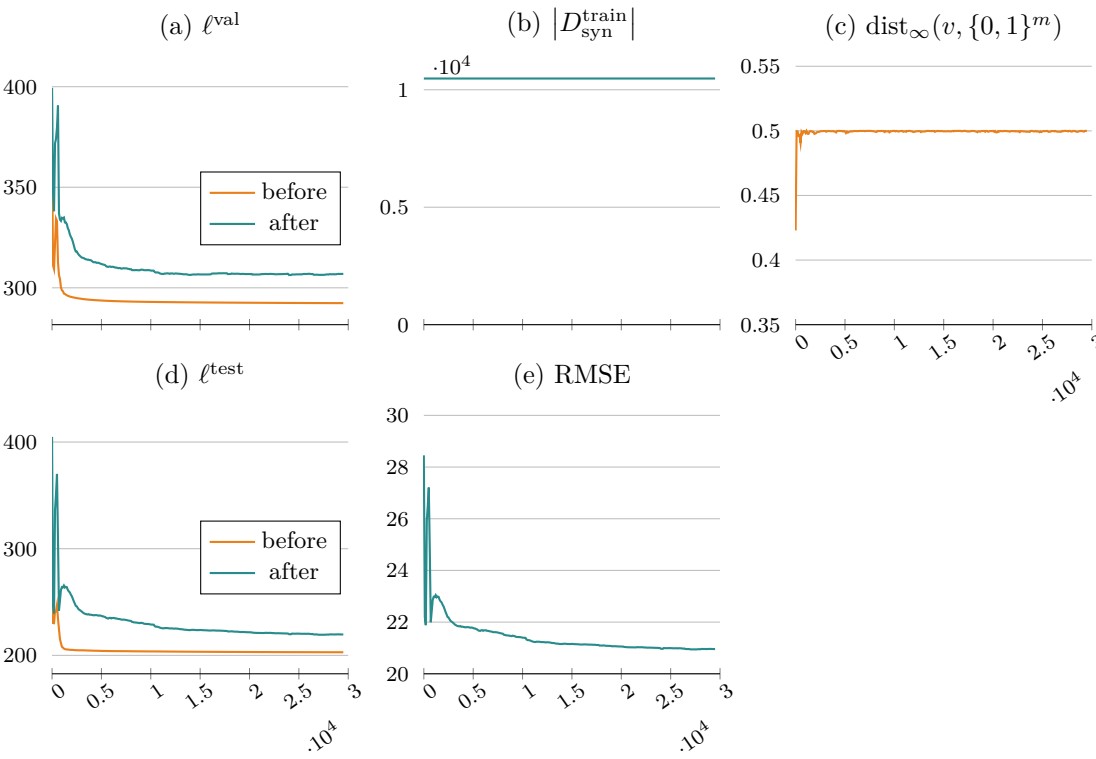

Figure 5: Evolution of the quantities reported in Table 3 for the *relaxation and rounding* method with *top-k hard thresholding* rounding on the *music* dataset with distillation budget at 20% of the training set size.

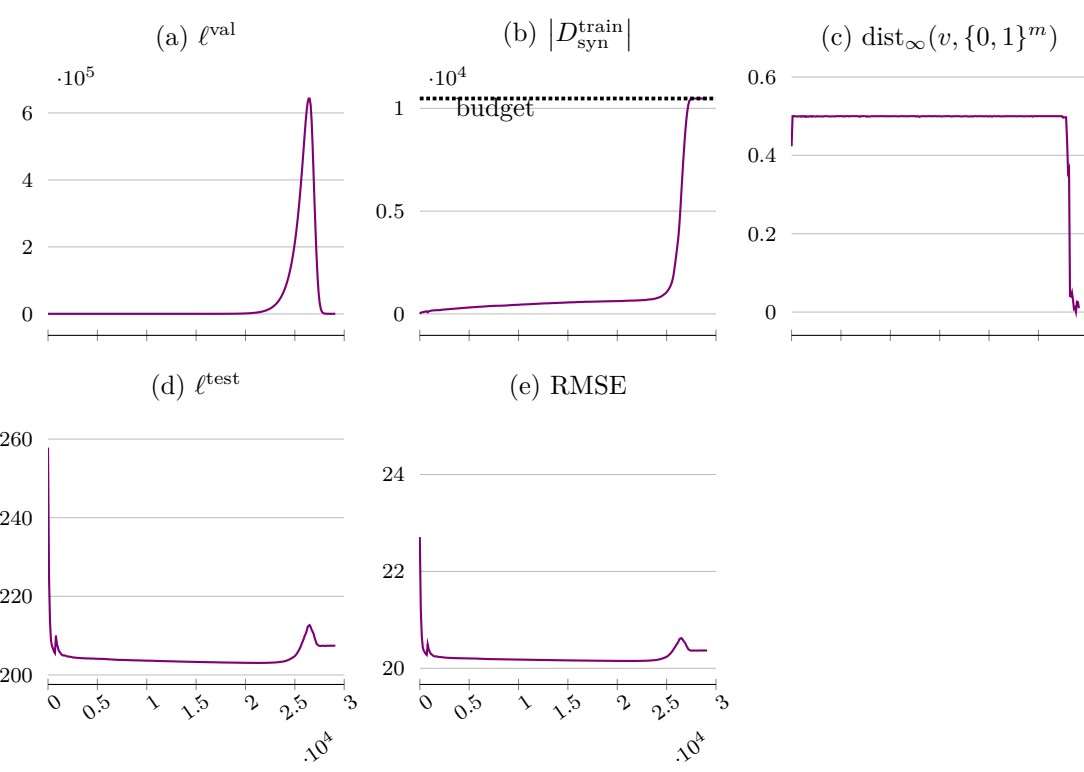

Figure 6: Evolution of the quantities reported in Table 3 for the *relax and penalize* method on the *music* dataset with distillation budget at 20% of the training set size.

