# OpenReview forum: "Relax and penalize: a new bilevel approach to mixed-binary hyperparameter optimization"
_TMLR — Accepted by TMLR_

### Review · Reviewer_9wEJ · 2024-12-02

**Summary Of Contributions:**

The authors propose a new approach for binary parameters in bilevel optimization.
Their approach is theoretically motivated and shows performance improvements in the case of reconstruction errors on two real datasets (data distillation problem) and in simulated regression context with group sparsity.

**Audience:**

Yes

**Broader Impact Concerns:**

No concerns

**Claims And Evidence:**

Yes

**Requested Changes:**

See weaknesses related to comments for:

- Algorithm 1 and $\beta$ hyperparameter
- Table 1-2 possible issue
- Add more discussion regarding the assumptions for theorem 3

### Typos and technicalities

- Misusage of `\citet` and `\citep` in the paper (almost everywhere), please correct it
- Use either `\emptyset` or `\varnothing` but not both to be consistent (between main paper and appendices)

**Strengths And Weaknesses:**

First I would like to express that this is not exactly my area of expertise, that said the paper is well written to still understand its motivation so thank you for that.

### Strengths

- The paper is well written and organized
- The method proposed is theoretically grounded
- The application show performance improvements

### Weaknesses

- One issue that I have is the $\beta$ hyperparameter presented in Algorithm 1 and only referred in the appendix by giving it's value for each experiment.
    - How were these values selected (grid search? If so what was the selection criteria, over what space)
    - No ablation / impact study is provided in the paper (what is the impact of the $\beta$, a high $\beta$ wil create more thresholds until convergence, leading longer running time -- and maybe finding other local minima?) This should be reported and discussed.
    - The $\beta$ is fixed, this seems "naive" as adaptative values could be chosen (the more penalized the problem, the faster the convergence and during runtime have a $\beta$ with higher values will have finer grains of precision)
    - The number of problems to solve in App. C.2 is set to K=6 and in D.3 K=10. Shouldn't the sequence of problems to solve not have a limited number of steps and only be stopped given a stoppping criterion (based on the convergence/stationnarity of the objective and the constraints satisfaction)? In this case I would expect a curve representing the evolution of the objective along the different steps k and in parallel the evolution of $\theta^k$'s violation of the binary constraint.

- Table 1-2: Maybe I did not understand it correctly, but shouldn't *relax and penalize* be refered as *p* and not *r* (and switch with the rounding method name) in captions/first column. Otherwise as it represents test error / reconstruction error (to minimze) it seems that relax and penalize performs worse than the rounding method. Please either correct it or discuss more the results obtained.

- The assumption for Theorem 3 that $d_\infty (\theta^k, \Theta_{\text{bin}})$, while being monitored (Remark 3) is briefly discussed but is also intriguing (not really a weak point more a curiosity). Has this condition ever been breached in the experiments in Section 5? If so, do you propose to simply project the obtained vector $\theta^k$ -- via rounding maybe to combine with existing work. Can you minitor the evolution of this quantity during optimization and provide the results ? This would be of interest to show that the assumption is indeed verified.

---

> ### Author Response · Authors · 2025-01-04
> **Summary**
>
> We thank the Reviewer 9wEJ for their thorough review of our manuscript.
> We submitted a preliminary revised version of the paper that addresses all the issues of Reviewer. All changes
> are indicated in blue throughout the manuscript.
>
> We would also like to emphasize that we are currently conducting the experiments requested by Reviewers on the group lasso problem (ablation study on the $\beta$ parameter and inclusion of the top-$k$ hard thresholding baseline).  We will add those results to the final version of the paper as soon as the new runs are completed.

---

> ### Author Response · Authors · 2025-01-04
> **Algorithm 1 and  $\beta$ hyperparameter**
>
> We thank the Reviewer for pointing out that the $\beta$ hyperparameter requires more careful analysis and explanation.
>
> - In the experiments, we selected the $\beta$ hyperparameter so that, starting from $\varepsilon^0$, we reach a suitably chosen $\varepsilon^{K}$ after $K$ iterations. More specifically, we set a sufficiently large $\varepsilon^0$ such that the penalty term does not dominate the function and a sufficiently small $\varepsilon^{K}$ such that the penalty term has more weight, encouraging integer solutions. We thus have the following expression:
> \begin{equation*}
> \beta = \sqrt[K]{\frac{\varepsilon^{K}}{\varepsilon^0}}.
> \end{equation*}
> However at the moment, in order to both better exploit the hyperparameter $\beta$ and properly address the issue raised by the Reviewer in point (b), we are running the experiment again choosing $\beta$ through a grid search in the range $[0.1,0.9]$ with step $0.2$.
>
> - For each experiment, we picked one setting to carry out the cross-validation across $\beta$. More specifically, for the group lasso problem we considered $a=0.3$, and for the data distillation problem, we picked $perc=20\%$. In the new version of the paper we included the ablation study of $\beta$ on data distillation (group lasso results will be included once the related code is done running).
>
> - We keep the $\beta$ fixed as common in penalty approaches in literature [1, 2].
> Furthermore, we wanted to avoid a costly multi-dimensional grid search, since one needs to cross-validate the $\beta$ for each internal iteration.
>
> - We apologize for not being clear in App. C.2 and D.3.
> The true stopping criterion is the condition $d_\infty(\theta^k, \{0,1\}^n)<tol$ with $tol=10^{-2}$ tolerance.
> In the applications, this condition is always achieved within $K=10$ iterations, which we used to set the hyperparameter $\beta$ as described in point (a). In the new experiment, since $\beta$ is chosen by cross-validation, we keep the same stopping criterion above, without knowing in advance the maximum number of external iterations $K$.
> In the new version (App. D.4), we report the evolution along the iterations of the binary constraint violation and the quantities reported in Table 3, for the $blog$ dataset with a distillation budget at $20\%$ of the training set size.
>
> [1] Stefano Lucidi and Francesco Rinaldi. An exact penalty global optimization approach for mixed-integer programming problems. Optimization Letters, 7:297–307, 2013.
>
> [2] Walter Murray and Kien-Ming Ng. An algorithm for nonlinear optimization problems with binary variables. Computational optimization and applications, 47(2):257–288, 2010.

---

> > ### Comment · Reviewer_9wEJ · 2025-01-07
> >
> > I greatly appreciate the added explanations and the (costly) ablation study that was added to the paper.
> > As you already answered to reviewer 6nSq, this let to a new value for the $\beta$ hyperparameter: answering my next question of why the results have changed so much since the first submission.
> >
> > I don't really think removing them was necessary, these experiments conducted could still me included in an appendix as they are numerical results that could be of value.
> >
> > And to finish, I do not have any further question. Thank you for your modifications

---

> ### Author Response · Authors · 2025-01-04
> **Table 1-2 possible issue and typos**
>
> We apologize for the confusion and thank the Reviewer for finding the typo. We correct it in the paper.

---

> ### Author Response · Authors · 2025-01-04
> **Add more discussion regarding the assumptions for theorem 3**
>
> We thank the Reviewer for this comment. In the revised version (Figure 2 in App. D.3), we insert plots related to the evolution of this quantity to better assess the behavior of the method.

---

> ### Author Response · Authors · 2025-01-06
> **Completion of the ablation study concerning the $\beta$ parameter for the group lasso**
>
> In the final version we included the ablation study on the $\beta$ parameter for the group lasso problem.
>
> We hope that this study together with the analogous one for the data distillation problem answer to the Reviewer's concerns.

---

### Review · Reviewer_6nSq · 2024-12-10

**Summary Of Contributions:**

The optimization of machine learning systems can be formulated as a bilevel optimization, where the outer level optimizes the hyperparameters and the inner level optimizes model parameters given specific hyperparameters. For settings with binary hyperparameters, the mixed-binary nature poses a challenge to the optimization. Prior work often relaxes the binary domain dimensions to the continuous range $[0, 1]$, solves the relaxed problem, and gets to a solution in the original mixed-binary domain based on the continuous solution. However, this two-stage approach lacks theoretical guarantees for optimality in the mixed-binary domain.

This paper introduces a novel approach called *relax and penalize*, which adds a quadratic penalty term to the bilevel objective and optimizes over the relaxed continuous domain. The penalty term encourages solutions close to the mixed-binary domain. Theoretical analysis show that under certain conditions, this penalized bilevel optimization over the relaxed continuous domain is equivalent to the original mixed-binary optimization in terms of global optimizers, and its local optimizers are in the mixed-binary domain under some additional conditions.

Based on the theoretical results, an iterative algorithm is proposed that applies an exponentially larger penalty weight to find local optimizers of mixed-binary bilevel optimization. The proposed method is evaluated against a two-stage approach that first solves the bilevel optimization over a relaxed continuous domain and then gets a binary solution by rounding each variable that should be binary. Empirical results show that the proposed method consistently outperforms this *continuous relaxation and rounding* approach across two machine learning tasks involving mixed-binary bilevel optimization.

**Audience:**

Yes

**Claims And Evidence:**

No

**Requested Changes:**

* As mentioned in the weaknesses discussion, a more appropriate baseline—such as the *continuous relaxation followed by Euclidean projection onto the feasible mixed-binary domain* method—is needed to provide a reasonable empirical comparison between the proposed method and prior work on the test tasks.

**Strengths And Weaknesses:**

Strengths:

* The proposed *relax and penalize* method introduces a novel approach to mixed-binary bilevel optimization. By incorporating a quadratic penalty term, the proposed penalized optimization—although formulated as a continuous relaxation—is theoretically proven to be equivalent to the original mixed-binary bilevel optimization (under certain conditions) in terms of global optimizers. Furthermore, additional analysis shows that local optimizers in the mixed-binary domain can be practically found by examining local optimizers of the penalized relaxation. This provides strong theoretical guarantees that are absent in the two-stage method used in prior work, which relaxes the bilevel objective to the continuous domain directly without a penalty term and then convert the continuous solution to a mixed-binary form.

* Hyperparameter optimization with mixed-binary constraints is a problem that has practical applications, as demonstrated by the two test tasks used in the empirical section of the paper. The method proposed in the paper makes a step further in the research of suitable methods for hyperparameter tuning of machine learning systems.

* The paper is overall well-written, with the proposed method and its theoretical contributions presented clearly and effectively. The remarks following the theorems are quite helpful for understanding the results themselves as well as the underlying motivations.

Weaknesses:
* My major concern about the paper is the mischaracterization of the methods in prior work and the choice of comparing against the *continuous relaxation and rounding* baseline:
  * The paper claims that "our method can outperform state-of-the-art approaches based on relaxation and rounding", and in Section 5.1 the baseline is presented as the method of Frecon et al., 2018 [1]. However, the *continuous relaxation and rounding* baseline, as defined in the paper, is not an accurate characterization of the two-stage methods used in the cited related works such as Frecon et al., 2018 and Zhang et al., 2022 [2]. After solving the relaxed continuous problem in the first stage, these two baselines obtain the final mixed-binary solution based on the continuous solution in a quite different way than the *continuous relaxation and rounding* baseline in the paper.
    * The *continuous relaxation and rounding* baseline in the paper obtains the mixed-binary solution by rounding each binary variable (hyperparameter) individually to 0 or 1.
    * For the group lasso task, Frecon et al., 2018 obtains the mixed-binary solution by assigning each feature to its most dominant group (See Section 4.1 of Frecon et al., 2018). A continuous solution of $[0.3, 0.4, 0.2, 0.1]$ would be rounded to $[0, 1, 0, 0]$, assuming these four variables represents the grouping weights for one feature. In contrast, the *continuous relaxation and rounding* baseline would round the solution to the all-zero vector $[0, 0, 0, 0]$, not assigning this feature to any group, essentially violating the feasibility constraint of the original grouping problem that $\sum_{l=1}^L \theta_{j,l}=1$ for every $j \in [P]$.
    * Zhang et al., 2022 obtains the mixed-integer solution by doing a top-k hard thresholding operation to the continuous solution, which can be equivalently defined as getting the Euclidean projection of the continuous solution onto the feasible mixed-integer domain (i.e., the domain that is defined by both the mixed-integer constraints and the feasibility constraints). (See discussions of *Euclidean projection* and *top-k hard thresholding* in Zhang et al., 2022.) This would be a more suitable approach for the data distillation task. For example, with $5$ training samples and a budget of $3$, the continuous solution $[0.4, 0.9, 0.4, 0.1, 0.2]$ would be projected to $[1, 1, 1, 0, 0]$ with the top-k hard thresholding. The *continuous relaxation and rounding* baseline would round this to $[0, 1, 0, 0, 0]$, under-utilizing the budget.
  * These differences between the prior work and the *continuous relaxation and rounding* baseline likely have a substantial impact on performance in the paper's test tasks. The *continuous relaxation and rounding* baseline is not suitable for these tasks as it produces excessive zero values in the rounded solution, leading to infeasibility (multiple features that are not assigned to any group) in Figure 1 and significant under-utilization of the knapsack budget in Table 3.
  * The *continuous relaxation followed by Euclidean projection onto the feasible mixed-binary domain* method as mentioned above would align with the prior works much better, and it does not require specific knowledge of each tasks's structure. This approach would be a more reasonable baseline and would likely performe significantly better than the current baseline in the test tasks.


[1] Jordan Frecon, Saverio Salzo, and Massimiliano Pontil. Bilevel learning of the group lasso structure. In Advances in Neural Information Processing Systems 31, pp. 8301–8311. Curran Associates, Inc., 2018.

[2] Yihua Zhang, Yuguang Yao, Parikshit Ram, Pu Zhao, Tianlong Chen, Mingyi Hong, Yanzhi Wang, and Sijia Liu. Advancing model pruning via bi-level optimization. In Advances in Neural Information Processing Systems, 2022.

---

> ### Comment · Reviewer_6nSq · 2025-01-04
>
> Dear Authors,
>
> This is a quick reminder that the two-week discussion period is ending soon. I raised a concern in my review regarding the implementation of the *continuous relaxation and rounding* method and its differences from the approaches used in baseline papers. I’d greatly appreciate hearing your thoughts on this matter before the discussion period concludes.

---

> ### Author Response · Authors · 2025-01-04
> ***Continuous relaxation followed by Euclidean projection onto the feasible mixed-binary domain* baseline**
>
> We thank the Reviewer for these comments. We agree that a comparison with the other suggested baselines would give better insights into the behavior of the proposed methodology. In the new version of the paper, we already included a comparison with the *relaxation and rounding* method that embeds the top-$k$ hard thresholding strategy suggested by the Reviewer. The results clearly show that the *relax and penalize* method outperforms *relaxation and rounding* with the *top-$k$ hard thresholding* strategy.
>
> However, we would like to apologize for the delay in providing the complete set of results. While we have made significant progress, the experiments for the group lasso problem are still ongoing (both the ablation study of the $\beta$ parameter requested by Reviewer 9wEJ and comparison on the new baseline). These tests require additional time due to a large number of runs needed. We are actively working to complete them as quickly as possible and will incorporate the results into the revised version of the paper once all tests are finalized.
>
> We deeply appreciate your understanding and patience as we finalize this work. Please let us know if you have any questions or require additional information in the meantime.

---

> > ### Comment · Reviewer_6nSq · 2025-01-05
> >
> > I appreciate the authors providing the experimental results for the *relaxation and rounding (top-k)* method on the data distillation task. This top-k rounding version offers a much more reasonable baseline for comparison than the *relaxation and rounding (simple)* version. The new results show that the proposed *relax and penalize* technique generally outperforms the *relaxation and rounding (top-k)* baseline, though the performance gap is narrow in some cases. That said, I have a few follow-up questions regarding the revised paper:
> >
> > * It seems the performance of the *relaxation and rounding (simple)* method in Table 3 of the revised paper is quite different from its performance in the corresponding experiments in the initial submission: *relaxation and rounding (simple)* in general utilizes much more budget and therefore has much better performance in the revised paper across the cases in Table 3. For example, in the case of *music (10%)*, the training set it constructs is of size 12,764 in the revised paper, but the size was only 514 in the initial submission. I wonder what experimental settings have changed to cause this notable difference in the simple rounding method.
> > * It also seems that there is an issue with the *relax and penalize* method in the case *blog (30%)*: its budget utilization is 52,397, but it should be 15,719 instead.
> >
> > As mentioned in the authors' comment above, the results for the *relaxation and rounding (top-k)* baseline on the group lasso task are yet to be added in the revised paper. I thank the authors for making the efforts to add these experiments, and I think having results on them would provide great value to the submission for the reasons discussed in my review.

---

> ### Author Response · Authors · 2025-01-05
>
> - Due to the ablation study on $\beta$, we now use $\beta=0.9$ for the data distillation problem, which implies that the proposed penalized method satisfies the stopping condition after a larger number of total iterations. Since we run the *relaxation and rounding* method with the same number of iterations, this now gives better results also for the *simple* baseline.
> -  We thank the Reviewer for noticing the typo in the table. We will fix it in the final version.
>
> We will soon upload the results for the group lasso problem.

---

> ### Author Response · Authors · 2025-01-06
> **Revision of the group lasso experiments**
>
> In the final version of the paper we have completed the numerical experiments related to the group lasso problem.
>
> We appreciate the Reviewer's suggestion concerning the top-k hard thresholding rounding.
>
> Overall, these preliminary experiments on the group lasso problem show that the proposed method and the relaxation and rounding method with top-1 hard thresholding perform essentially the same, and they are both preferable to the relaxed and simple
> rounding procedure. We believe that the effectiveness of the top-1 hard thresholding is mainly due the the fact that we
> are considering nonoverlapping group structures which put strong constraints on the binary variables, since it is all
> about identifying one nonzero entry for each feature.
>
> Furthermore, we would like to highlight that this study (together with the ablation study on the parameter $\beta$ required by Reviewer 9wEJ) was particularly time consuming due to the large number of synthetic group lasso problems involved in the tests. To make the analysis feasible within the available time, we decided to
>
>  - focus on a specific scenario with $a = 0.3$ as a representative case in the ablation study on the $\beta$ parameter;
>
> - address the settings with $a=0.1,0.3,0.5$ in the comparison against the new baseline and perform only 3 different runs for each scenario.
>
> This allowed us to rerun all tests for the two methods under consideration, namely relaxation and rounding and relax and penalize.
>
> We remain available to address any further questions or concerns the Reviewer may have.

---

### Review · Reviewer_Sbp8 · 2024-12-23

**Summary Of Contributions:**

The authors consider bilevel optimization where some of the leader variables are binary (the follower variables are continuous), under the assumption that the follower problem has a unique solution. This is a quite general problem with a lot of interesting real applications; the authors focus on machine learning problems that show bilevel structure.

The authors start by focusing on the leader’s problem only (leaving the follower’s best-response map implicit) and replacing the binary constraints with a penalty term from the literature.

Their main new result is that is that, for this penalty problem, and for small enough penalty term, all *local* minimizers that are sufficiently close (in $\ell_\infty$ distance) to a binary feasible point, will have to be binary feasible. (In their experiments, they actually check that the assumptions for this do hold.) They also extend old results showing that for small enough penalty term, the penalized problem’s global minimizer will be correct for the original problem.

This provides motivation for their simple but effective optimization algorithm. They gradually increase the penalty term, locally solving the penalized problem at each point. At some point, they will end up with a relaxed problem whose global minimizer is correct. In practice, this is nice but not directly relevant, because for nonconvex optimization, one can only really hope to find a local minimizer, but their new result at least guarantees that eventually, their local minimizers will be binary-feasible. Experiments are on two reasonable and well-studied (though slightly niche) machine learning tasks and show it works well compared to defensible baselines.

**Audience:**

Yes

**Broader Impact Concerns:**

No broader impact concerns.

**Claims And Evidence:**

Yes

**Requested Changes:**

No requested changes.

**Strengths And Weaknesses:**

Strengths:

- The analysis is done carefully. The authors are very clear about what assumptions have to hold in terms of smoothness, uniqueness of follower solutions, proximity of local solutions to binary feasibility, etc. and these assumptions are reasonable for the experimental settings they consider.
- The method appears to work well on interesting machine learning problems.
- Bilevel optimization (especially with some discrete variables) is fairly important and general, and there are probably broader applications of this technique.

Weaknesses:

- There are no substantial weaknesses to this paper that I can see.

---

> ### Author Response · Authors · 2025-01-04
>
> We thank the Reviewer for the positive feedback. We appreciate the recognition of the clarity and rigor in the analysis, as well as the support for the potential applications of the presented approach in machine learning.

---

### Author Response · Authors · 2025-01-06
**Final Comments**

We would like to thank again all the Reviewers for carefully reading our paper. We deeply appreciated their useful suggestions.

We hope that the final version of the paper answers to their concerns. We remain available to address any further  questions or concerns.

We finally thank the Action Editor for handling our submission.

---

### Decision · Action_Editor_V6c4 · 2025-01-29

**Recommendation:** Accept with minor revision

**Comment:**

Based on the back-and-forth with the reviewers and the revisions made, I am very happy to say this paper is now a quite clear accept, with contributions that are well supported.

There are a number of very minor comments that should be addressed before the camera ready:

 - Quoting from a reviewer: "In the latest version, the summaries of the empirical results in the introduction and conclusion sections have not been updated to accurately reflect the new experimental results added since the initial submission. For example, the conclusion currently suggests that the proposed method outperforms the relaxation and rounding baseline on both the group lasso problem and the data distillation task. This was the case for the old experiments in the initial submission which had a different baseline implementation, but with the latest experiments, the proposed method shows better performance only on the data distillation task. These summaries in the paper should be revised to align with the updated experimental results."
 - Minor formatting issues:
    - Tables have inconsistent font sizes; if small font is needed in a table, please use small font for all tables.
    - There is an extra space before the period at the end of Table 3's caption
    - Some Remarks that have subpoints (i) and (ii) begin with a line break but others don't. I recommend adding the line break so that (i) and (ii) line up neatly.
    - In Table 1 and 2 and maybe elsewhere you have some instances of $\theta^{top}$ which should be styled as $\theta^{\text{top}}$ using a \text environment, otherwise the letters "top" are kerned incorrectly as if they were a product of one-character variables. This might also be the case for "music", "blog", and "perc" in Table 3, where probably \emph is more appropriate if needed.

**Audience:**

The topic of mixed-binary optimization is one with many applications as well as theoretical interest, and the paper is a valuable contribution that should be communicated.

**Claims And Evidence:**

Reviewers agree that the paper makes interesting theoretical and empirical claims that are, after some revisions, well supported.